# TimeSpot: Benchmarking Geo-Temporal Understanding in Vision–Language Models in Real-World Settings

## Abstract

Geo-temporal understanding, the ability to identify the location, time, and contextual features of an image from visual cues alone, is a fundamental aspect of human intelligence with wide-ranging applications, from disaster response to autonomous navigation and geography education. While recent vision–language models (VLMs) have shown progress in image geo-localization using conspicuous cues like landmarks or road signs, their ability to understand temporal signals and related spatial reasoning cues remains underexplored. To address this gap, we introduce **TimeSpot**, a comprehensive benchmark for evaluating real-world geo-temporal reasoning in VLMs. TimeSpot comprises 1,455 images spanning 80 countries, where models must infer temporal attributes (season, month, time of day, daylight phase) and geolocation attributes (continent, country, climate zone, environment type, latitude–longitude coordinates) directly from the visual input. In addition, it includes spatial reasoning tasks that require integrating geographical, spatial, and temporal cues to solve complex understanding problems. Unlike prior benchmarks that emphasize obvious cues or iconic imagery, TimeSpot prioritizes diverse and subtle settings, reflecting the difficulty of reasoning under real-world uncertainty. Our evaluation of state-of-the-art VLMs, including both open- and closed-source models, reveals consistently low performance across tasks, highlighting substantial challenges in achieving robust temporal and geographic reasoning. These findings underscore the pressing need for improved methods to enable reliable and trustworthy geo-temporal understanding in VLMs, paving the way for future research in this critical domain.

## 1 Introduction

Determining *where* and *when* a photograph was captured using visual information alone is a fundamental human cognitive skill, underpinning situational awareness, episodic memory, and contextual reasoning. This capability requires the integration of diverse and often subtle visual cues, including illumination and shadow geometry, seasonal vegetation patterns, architectural styles and materials, clothing and traffic conventions, as well as broader geographic regularities in the natural and built environment (Lin et al., 2013; Workman et al., 2015b; Arandjelovic et al., 2016; Hu et al., 2018; Hu & Lee, 2020). We refer to this integrated ability as geo-temporal understanding. Beyond its cognitive significance, geo-temporal reasoning is central to high-impact applications such as disaster response and recovery (Mirowski et al., 2018), environmental and climate monitoring (Zhai et al., 2017), autonomous navigation and localization (Lynen et al., 2020; Sarlin et al., 2019), and media forensics and verification (Tian et al., 2017).

In the geospatial domain, substantial progress has been achieved through cross-view and street-view localization benchmarks (Vo & Hays, 2016). Early work on ground-to-aerial matching has evolved into large-scale, geographically diverse datasets and unified embedding frameworks, including VIGOR (Zhu et al., 2021), GeoCLIP (Vivanco Cepeda et al., 2023), OpenStreetView 5M (Astruc et al., 2024), Global Streetscapes (Hou et al., 2024), CV-Cities (Huang et al., 2024), panoramic cross-view settings (Xia et al., 2025), and recent embedding advances (Cai et al., 2025). Despite this progress, existing benchmarks almost exclusively focus on where, typically measured via retrieval ranks or coordinate error (Hu & Lee, 2020), while largely ignoring when. Explicit temporal inference

(e.g., season, month, or local time), as well as internal geo-temporal consistency (e.g., avoiding "July" paired with winter conditions in the Northern Hemisphere), is rarely required.

When temporal information is considered, it is usually treated indirectly or through proxy tasks such as navigation (Mirowski et al., 2018; Lynen et al., 2020), place recognition (Sarlin et al., 2019), or change detection (Sarlin et al., 2024), none of which demand structured temporal outputs. Recent geospatial vision–language benchmarks in remote sensing, including HRVQA (Li et al., 2024b) and GEOBench-VLM (Danish et al., 2024), emphasize aerial imagery and focus on classification, counting, or segmentation, rather than fine-grained temporal estimation or comprehensive geographic characterization. Moreover, these benchmarks do not require models to reason jointly over structured geographic attributes, such as continent, country, climate zone, environment type, and coordinates, from ground-level imagery. As a result, the community lacks a unified evaluation framework that probes joint geo-temporal reasoning, emphasizes subtle non-iconic cues over landmarks or text, and incorporates trustworthiness criteria such as schema validity, calibration, and robustness under distribution shift (Astruc et al., 2024; Hou et al., 2024; Huang et al., 2024; Xia et al., 2025).

This gap is consequential for three key reasons. **First**, temporal cues are often essential for disambiguation: solar geometry varies systematically with hemisphere (Ye et al., 2024), vegetation phenology separates climates at similar latitudes (Wang et al., 2024c), and daylight phases shape urban lighting and activity patterns (Shi et al., 2020; Zhu et al., 2022). **Second**, real-world deployment requires *structured, verifiable predictions*, including minute-level precision for local time, kilometer-scale tolerance for coordinates, and explicit cross-field consistency checks (e.g., rejecting "snow in July" for a Northern Hemisphere prediction). Retrieval-centric protocols lack such safeguards, encouraging over-reliance on superficial or spurious cues. **Third**, robust global generalization demands *diversity-aware stress testing*. Hemisphere inversions, climate-region shifts, and out-of-distribution (OOD) splits expose brittle heuristics and dataset biases that iconic or text-dependent benchmarks fail to reveal (Astruc et al., 2024).

To address these limitations, we introduce TIMESPOT, a benchmark for evaluating real-world geo-temporal understanding in vision–language models, illustrated in Figure 1. TIMESPOT comprises 1,455 natural, non-iconic images spanning 80 countries, curated to minimize landmark dependence and foreground low-salience visual evidence. For each image, models are required to predict a *structured schema* consisting of four temporal attributes: season, month, local time, and daylight phase, and five geographic attributes: continent, country, climate zone, environment type, latitude, and longitude. This design explicitly encourages models to integrate illumination cues, material properties, natural context, and human activity patterns, rather than exploiting textual artifacts or iconic structures. Beyond per-field predictions, TIMESPOT also includes *fusion questions* that require coordinated spatial–temporal reasoning, reflecting real-world scenarios in which time and place are inherently entangled.

Our contributions are threefold:

1. **A challenging joint geo-temporal benchmark.** TIMESPOT is, to our knowledge, the first benchmark to require simultaneous prediction of four temporal and five geographic attributes from natural images, explicitly targeting fine-grained spatiotemporal reasoning beyond conventional retrieval or classification.

2. **Rigorous and verifiable evaluation of modern VLMs.** We assess a diverse set of open- and closed-source models under uniform, metadata-free, and fully open-ended question–answering conditions. Our evaluation enforces precise metrics for temporal accuracy and geodesic error, explicit schema-consistency constraints (e.g., month–season–hemisphere alignment and climate plausibility), and calibration analysis. The results reveal low temporal accuracy and frequent geo-temporal inconsistencies, underscoring that reliable joint reasoning remains an open challenge.

3. **Comprehensive error analysis and root-cause investigation.** We conduct systematic analyses of model failures, identify recurring spatial and temporal error modes, and outline concrete mitigation strategies to guide future research.

Our evaluation framework is *verifiable*, enforcing window-based accuracy for temporal predictions and mean or thresholded geodesic distance for coordinates; *constrained*, requiring internal schema consistency across predicted fields; *robust*, incorporating hard and OOD splits without auxiliary metadata; and *calibrated*, assessing expected calibration error and risk–coverage trade-offs. When applied to state-of-the-art open- and closed-source vision–language models, the results expose

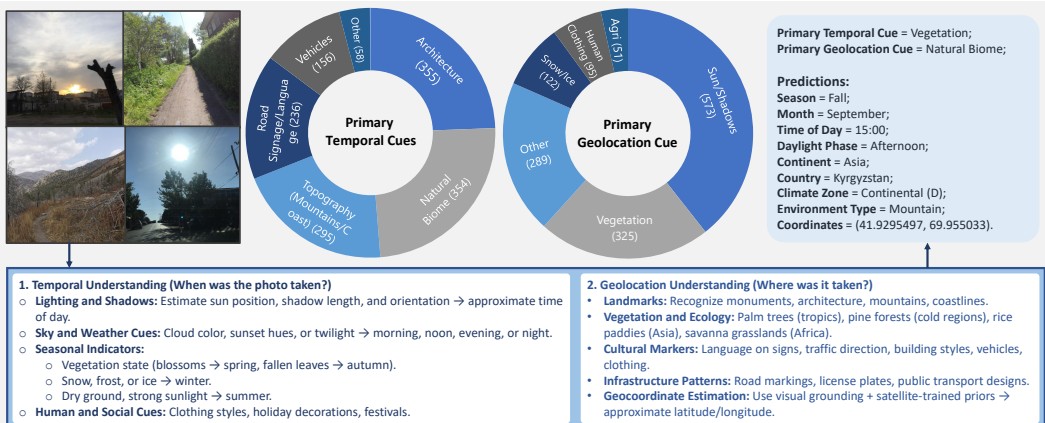

**Figure 1: Illustration of the TIMESPOT benchmark for geo-temporal understanding.** Models must infer temporal attributes (season, month, time of day, daylight phase) and geographic attributes (continent, country, climate zone, environment type, coordinates) directly from visual input. Left: example images. Center: distributions of primary temporal cues (e.g., architecture, natural biome, topography) and geolocation cues (e.g., sun/shadows, vegetation, snow/ice). Right: an example prediction, highlighting the integration of diverse and subtle cues required for reliable reasoning.

substantial deficiencies in joint geo-temporal understanding. Even the strongest models achieve 77.59% country accuracy (Gemini 2.5, Flash, Thinking) yet suffer a median geodesic error of 892.54 km, indicating reliance on coarse geographic heuristics. Temporal reasoning is even more fragile: time-of-day prediction peaks at only 33.74% accuracy (GLM 4.1V, 9B, Thinking), revealing weak grounding in illumination and contextual cues.

TIMESPOT complements existing geolocation and remote-sensing benchmarks by offering a unified, diversity-aware, and trust-focused evaluation of time and place. Our findings demonstrate that authentic geo-temporal understanding remains unsolved and highlight urgent avenues for improving robustness, calibration, and real-world applicability in next-generation vision–language models.

## 2 PRELIMINARIES AND RELATED WORK

TIMESPOT addresses geo-temporal inference in *ground-level visual scenes*, where explicit textual cues and iconic landmarks are deliberately minimized, and successful prediction requires fusing weak, spatially distributed signals. Given an image, a model must produce a structured output schema comprising four temporal attributes: season, month, local time (HH:MM), and daylight phase, and five geographic attributes: continent, country, climate zone, environment type, and latitude–longitude coordinates. This formulation shifts evaluation away from retrieval-based objectives (Vivanco Cepeda et al., 2023; Astruc et al., 2024; Hou et al., 2024; Huang et al., 2024) toward *interpretable and verifiable field-level predictions*.

**Formal Definition.** Formally, each image $x \in \mathcal{X}$ exposes observable cues such as illumination patterns, vegetation and phenological signals, material and architectural styles, and human activity traces. These cues map to a structured label $y = (y^{\text{temp}}, y^{\text{geo}})$, where the temporal component is $y^{\text{temp}} = (s, m, \tau, \phi)$ for season, month, local time, and daylight phase, and the geographic component is $y^{\text{geo}} = (C, \kappa, z, e, (\lambda, \varphi))$ for continent, country, climate zone, environment type, and coordinates. The evaluation task probes a fixed vision–language model through a mapping $f : \mathcal{X} \mapsto \widehat{\mathcal{Y}}$, where $\widehat{\mathcal{Y}}$ denotes predictions in the same structured schema. This setting explicitly evaluates abilities that generic VLMs (Radford et al., 2021; Li et al., 2022; Kim et al., 2021) and remote-sensing benchmarks do not directly test: interpreting solar geometry, phenological states, material cues, and contextual regularities to jointly localize scenes in both space and time. The geographic coverage of TIMESPOT is shown in Figure 2.

**Relation to General Multimodal Benchmarks.** General-purpose multimodal benchmarks emphasize broad perceptual and reasoning skills across images, text, charts, and diagrams, but provide limited assessment of *geo-temporal competence*, as per Table 1. Datasets such as MMMU (Yue et al.,

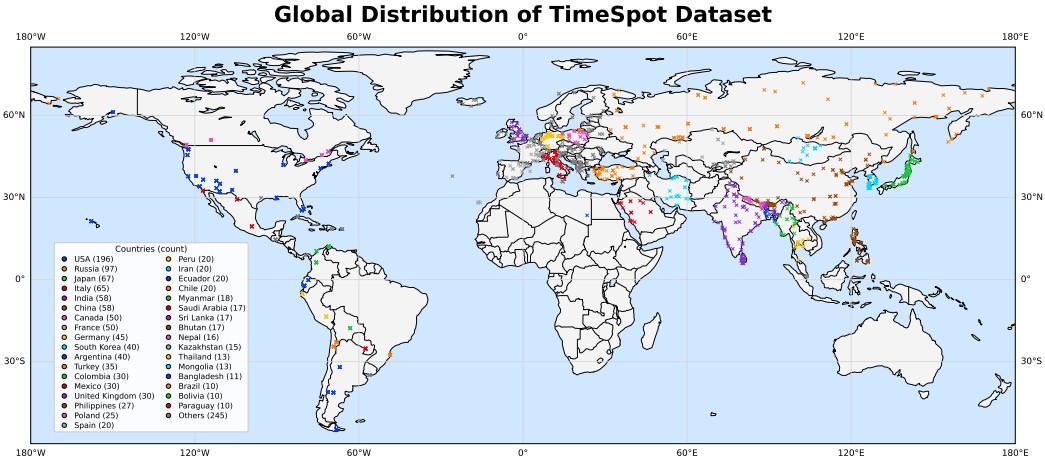

**Figure 2: Global coverage of TIMESPOT.** Locations of all 1,455 ground, level photos across 80 countries. Each marker denotes an image coordinate; colors indicate country (top contributors listed; "Others" aggregates the remainder). The dataset spans all inhabited continents and diverse climate zones and environments, providing non, iconic scenes for evaluating geo-temporal reasoning.

2024), M3Exam (Zhang et al., 2023), M4U (Wang et al., 2024a), MM-Vet (Yu et al., 2023), MME (Fu et al., 2023), MMBench (Liu et al., 2025), MMSTAR/Sphinx (Lin et al., 2023), SEED-Bench (Li et al., 2023), and SEED-Bench-2 (Li et al., 2024a) evaluate diverse capabilities ranging from scientific reasoning to map and chart understanding. However, they neither require explicit prediction of when and where a scene occurs nor enforce cross-field geographic and temporal validity constraints. Foundational vision–language pretraining efforts, such as CLIP (Radford et al., 2021), ViT (Dosovitskiy et al., 2020), ViLT (Kim et al., 2021), BLIP (Li et al., 2022), MGP/SigLIP (Zhai et al., 2023), and Long-CLIP (Zhang et al., 2025), provide strong visual and textual priors, yet remain largely agnostic to the physical and ecological principles that govern temporal and geographic context.

**Recent Works on Geolocation Understanding.** Recent work has substantially advanced image-based geolocation using large vision–language models, but has largely focused on *spatial inference alone*. LLMGeo (Wang et al., 2024d) and IMAGEO-Bench (Li et al., 2025) benchmark country-, city-, or coordinate-level localization with sophisticated prompting and structured reasoning, while ETHAN (Liu et al., 2024b) and agent-based evaluations (Jay et al., 2025) demonstrate that chain-of-thought strategies and navigation tools can significantly improve spatial accuracy. FAIRLOCATOR (Huang et al., 2025) further reveals systematic socio-economic and regional biases in spatial predictions, highlighting important fairness concerns. However, none of these benchmarks require predicting explicit temporal attributes (e.g., month, local time, daylight phase) or enforce physical geo–temporal consistency checks such as month–season–hemisphere or time–daylight compatibility. **TIMESPOT** is complementary to this line of work: assuming that modern VLMs already exhibit strong spatial recognition, it evaluates whether models can *jointly infer time and place* from subtle physical cues and maintain consistency across a nine-field geo–temporal schema. Our results show that even state-of-the-art models that perform well on spatial benchmarks exhibit large temporal errors and frequent geo–temporal inconsistencies, exposing a critical and previously unmeasured failure mode.

**Evaluation and Trustworthiness.** TIMESPOT evaluates categorical fields using top-1 accuracy, local time using minute-window accuracy and mean absolute error (MAE), and geographic coordinates using great-circle distance (mean, median, and thresholded ranges), following established geolocation protocols (Tian et al., 2017; Vo & Hays, 2016; Arandjelovic et al., 2016) while extending them to explicitly temporal prediction. To encourage *trustworthy inference*, we incorporate cross-field consistency constraints and diagnostics, including hemisphere–season agreement, daylight-phase compatibility with predicted time and location, and climate plausibility at (`lat`, `lon`), building on practices from large-scale cross-view and geolocation benchmarks (Zhu et al., 2021; Astruc et al., 2024; Hou et al., 2024). We further report calibration metrics, including expected calibration error (ECE) and risk–coverage curves, to assess confidence reliability in multi-field outputs, addressing failure modes observed in prior cross-view and place-recognition systems (Lin et al., 2013; Workman et al., 2015b; Hu & Lee, 2020), where predictions can be confident yet physically inconsistent.

**Table 1: Comparison along TIMESPOT axes.** *Temp*: Season/Month/Time/Daylight phase. *FineGeo*: Continent/Country/Climate/Environment/Lat–Lon. *Subtle*: non-iconic cue emphasis. *HS/OOD*: hemisphere sanity or hard/OOD splits. *FusionQs*: geo-temporal fusion tasks. *Schema*: structured field outputs. *Calib*: calibration/uncertainty metrics. *Verif*: GPS/OSM-verifiable scoring. *Globality*: multi-continental coverage. Symbols: ✓ = explicit support; △ = partial/limited; — = not present.

| Benchmark / Dataset (year) | Temp | FineGeo | Subtle | HS/OOD | FusionQs | Schema | Calib | Verif | Globality |
|---|---|---|---|---|---|---|---|---|---|
| OpenStreetView–5M Astruc et al. (2024) | – | △ | △ | ✓ | – | – | – | ✓ | ✓ |
| Global Streetscapes Hou et al. (2024) | △ | △ | – | △ | – | – | – | ✓ | ✓ |
| CV–Cities Huang et al. (2024) | – | △ | – | ✓ | – | – | – | ✓ | ✓ |
| VIGOR Zhu et al. (2021) | – | △ | – | ✓ | – | – | – | ✓ | △ |
| CVACT Liu & Li (2019) | – | △ | – | ✓ | – | – | – | ✓ | △ |
| CVUSA Workman et al. (2015a) | – | △ | – | ✓ | – | – | – | ✓ | – |
| University–1652 Zheng et al. (2020) | – | △ | – | △ | – | – | – | ✓ | – |
| GeoText–1652 Chu et al. (2024) | – | △ | △ | ✓ | – | △ | – | ✓ | – |
| GeoCLIP Eval Vivanco Cepeda et al. (2023) | – | △ | – | ✓ | – | – | – | ✓ | ✓ |
| Panoramic Cross–View Xia et al. (2025) | – | △ | – | ✓ | – | – | – | ✓ | ✓ |
| HRVQA Li et al. (2024b) | – | – | – | △ | △ | – | – | – | ✓ |
| VRSBench Li et al. (2024c) | – | – | – | ✓ | △ | – | – | – | ✓ |
| GEOBench–VLM Danish et al. (2024) | △ | – | – | ✓ | △ | △ | – | – | ✓ |
| EarthVQA Wang et al. (2024b) | – | – | – | △ | △ | – | – | – | ✓ |
| FIT–RSFG / RSRC Luo et al. (2024) | – | – | – | △ | △ | △ | – | – | ✓ |
| RemoteCount Liu et al. (2024a) | – | – | – | △ | △ | – | – | – | ✓ |
| SkyEyeGPT / SkyBench Zhan et al. (2025) | – | – | – | △ | △ | – | – | – | ✓ |
| EO–VLM Benchmark Zhang & Wang (2024) | – | – | – | △ | △ | – | – | – | ✓ |
| RS5M Zhang et al. (2024) | – | – | – | – | – | – | – | – | ✓ |
| MapBench Hao et al. (2024) | – | – | – | ✓ | △ | – | – | ✓ | – |
| MapQA Chang et al. (2022) | – | – | – | △ | △ | △ | – | – | – |
| MapIQ Srivastava et al. (2025) | – | – | – | △ | △ | △ | – | – | – |
| CulturalVQA Nayak et al. (2024) | △ | △ | △ | △ | △ | – | – | – | ✓ |
| AMOS Time–lapse Jacobs et al. (2009) | △ | – | – | – | – | – | – | – | ✓ |
| Transient Attributes Laffont et al. (2014) | △ | – | – | – | – | – | – | – | ✓ |
| Transient Attributes Laffont et al. (2014) | △ | – | – | – | – | – | – | – | ✓ |
| LLMGeo Wang et al. (2024d) | – | ✓ | △ | – | – | – | – | △ | ✓ |
| ETHAN Liu et al. (2024b) | – | ✓ | △ | – | – | △ | – | △ | ✓ |
| Geo Inference Jay et al. (2025) | – | ✓ | – | – | – | – | – | △ | ✓ |
| FAIRLOCATOR Huang et al. (2025) | – | ✓ | △ | △ | – | – | – | △ | ✓ |
| IMAGEO-Bench Li et al. (2025) | – | ✓ | △ | – | – | ✓ | △ | △ | ✓ |
| **TimeSpot (Ours, 2025)** | ✓ | ✓ | ✓ | ✓ | ✓ | ✓ | ✓ | ✓ | ✓ |

**Generalization and Robustness.** To evaluate robustness, TIMESPOT provides stratified analysis across continents, climate zones, and environment types, alongside *hemisphere-flip* tests and hard out-of-distribution (OOD) splits that suppress landmark shortcuts and amplify reliance on physical and ecological cues (Astruc et al., 2024; Hou et al., 2024; Huang et al., 2024; Xia et al., 2025). The benchmark also includes fusion questions that require coherent integration of spatial and temporal evidence, exposing independence assumptions that commonly arise in modular VLM decoders (Lin et al., 2013; Hu et al., 2018; Zhu et al., 2021). Relative to generic multimodal evaluations and remote-sensing-focused geospatial suites, TIMESPOT occupies a distinct and currently missing niche: a *ground-level*, *joint geo-temporal* benchmark with structured outputs, internal consistency checks, calibration analysis, and systematic robustness diagnostics. Empirically, we observe that state-of-the-art open and closed VLMs, despite strong performance on broad multimodal benchmarks (Vivanco Cepeda et al., 2023) and cross-view geolocation tasks (Lin et al., 2013; Workman et al., 2015b; Hu & Lee, 2020), exhibit low temporal accuracy and frequent geo-temporal inconsistencies, highlighting substantial remaining headroom for physically grounded visual reasoning.

Additional related work is provided in Appendix A.

## 3 TIMESPOT BENCHMARK

TIMESPOT is a benchmark for evaluating *geo-temporal reasoning* in vision–language models (VLMs) from natural, non-iconic, ground-level imagery. The benchmark is designed to probe inference from subtle, distributed physical cues—such as illumination and shadow geometry, sky conditions, vegetation phenology, architectural materials, and human activity patterns—rather than from iconic landmarks or textual artifacts. It comprises 1,455 photographs spanning 80 countries and requires models to predict a structured schema with nine fields: four temporal attributes (*season, month, local time (HH:MM), daylight phase*) and five geographic attributes (*continent, country, climate*

*zone*, *environment type*, and *latitude, longitude*). By jointly requiring spatial and temporal inference, TIMESPOT extends prior geolocation evaluations beyond retrieval accuracy and coordinate error (Tian et al., 2017; Vo & Hays, 2016; Arandjelovic et al., 2016) and targets physically grounded reasoning under real-world uncertainty. Dataset statistics are summarized in Table 2.

## 3.1 BENCHMARK CONSTRUCTION

To ensure reproducibility while limiting annotation bias, TIMESPOT adopts a hybrid labeling strategy that combines deterministic programmatic derivation with structured human verification. This design scales annotation while preserving semantic validity in ambiguous cases where visual evidence, metadata, and physical modeling must be reconciled.

**Image collection and curation.** Images were collected from a combination of public web sources and manually captured photographs, with deliberate suppression of landmark-dominated scenes and text-rich imagery. Curation prioritized scenes where inference depends on synthesizing fine-grained physical evidence, such as distinguishing seasonal vegetation states, estimating solar elevation from shadow vectors, or inferring climate from vegetation and built environment. Sampling was explicitly balanced across hemispheres, latitude bands, climate zones, and environment types to ensure global diversity while minimizing memorization shortcuts from iconic locations.

**Programmatic label derivation.** Ground-truth labels were derived deterministically from capture metadata and geographic priors. Months were obtained from timestamps; seasons were assigned using meteorological definitions with hemisphere correction; daylight phases were computed from solar elevation and azimuth using civil, nautical, and astronomical twilight thresholds; and local time was derived from time zones and ephemerides. Climate zones were mapped via a global Köppen–Geiger classification (Peel et al., 2007); continent and country were assigned from coordinates; and latitude and longitude were retained as continuous targets. This procedure guarantees reproducibility and enforces physical consistency across temporal and geographic fields.

**Human verification.** All labels undergo a two-stage verification process. Annotators first gather auxiliary geographic context (e.g., locality, proximity to water, road configuration, and Street View imagery when available) and cross-check programmatic labels against visual cues such as illumination, shadows, vegetation, and weather conditions. A senior annotator then adjudicates edge cases—including sunrise/sunset ambiguity and artificial lighting—by validating metadata against ephemeris calculations and scene-level consistency.

**Schema and normalization.** Ground-truth annotations are stored in a canonical JSON schema. Model outputs are normalized to enable exact per-field scoring (e.g., "Autumn" → "Fall", signed decimal degrees for coordinates). Automated integrity checks enforce month–season–hemisphere consistency, daylight-phase compatibility with predicted time and location, and climate plausibility at (`lat`, `lon`), yielding auditable error modes beyond simple retrieval failure.

## 3.2 ANNOTATION AND QUALITY CONTROL

Unlike semantic labeling tasks where inter-annotator agreement serves as a proxy for reliability, TIMESPOT targets objective physical quantities deterministically derived from capture metadata and geospatial databases. Temporal fields are computed from timestamps and solar ephemerides, while geographic fields are directly mapped from GPS coordinates. Human annotators function exclusively as verifiers rather than labelers, auditing images to identify and reject cases with corrupted metadata or inconsistent visual evidence. This ensures that every retained sample maintains ground-truth validity grounded in verifiable physical data. The verification pipeline consists of:

(i) *Context collection*, where annotators gather map-level evidence and note relevant features impacting temporal cues (e.g., horizon occlusion, canyon geometry);

(ii) *Verification*, where programmatic labels are cross-checked against the image under formalized guidelines covering seasonality, hemisphere rules, solar elevation, twilight tolerances, and environment categorization—low-confidence samples are flagged;

(iii) *Expert temporal review*, where senior annotators adjudicate flagged cases by comparing ephemerides with observed shadow vectors and occlusion; deterministic temporal labels remain tied to metadata unless documented errors exist, while non-deterministic fields (e.g., environment type) may be revised with justification;

**Table 2: Dataset statistics.** We report coverage across temporal, geographic, cue, and environment axes.

| Axis | Field | Categories (top shown) |
|---|---|---|
| Temporal | Season | Summer (400), Fall (399), Spring (335), Winter (321) |
| | Daylight phase | Afternoon (584), Night (287), Sunset (210), Morning (203), Midday (124), Sunrise (47) |
| | Month | 12 months represented; top: August (163), September (146), July (145), March (131) |
| | Hemispheric tag | Northern Hemisphere Summer (703), Northern Hemisphere Winter (615), Southern Hemisphere Winter (81), Southern Hemisphere Summer (56) |
| | Time coverage | Day (1182), Night (273) |
| | Hour range | Full 0–23; densest 08–18 |
| Geography | Continents | Asia (529), Europe (430), North America (326), South America (170) |
| | Countries | 82 unique; top: USA (196), Russia (97), Japan (67), Italy (65), China (58) |
| | Climate | Temperate (C) (582), Continental (D) (396), Tropical (A) (274), Arid (B) (180), Polar (E) (23) |
| | Environment type | Urban (648), Rural (202), Mountain (193), Coastal (181), Suburban (118), Desert (113) |
| | Lat/Lon span | lat -54.80 to 71.96, lon -173.24 to 170.31 |
| Cues | Primary temporal cues | Sun/Shadows (573), Vegetation (325), Other (289), Snow/Ice (122), Human Clothing (95), Agricultural Activity (51) |
| | Primary geolocation cues | Architecture (355), Natural Biome (354), Topography (Mountains/Coast) (295), Road Signage/Language (236), Vehicles (156), Other (58) |

(iv) *Constraint audit*, where machine-enforced integrity checks on month–season–hemisphere alignment, phase–time consistency, and climate plausibility finalize versioned JSON records with annotations, decisions, and validation notes.

## 3.3 EVALUATION

Our schema supports *interpretable* predictions and *verifiable* evaluation. Categorical fields are scored with top-1 accuracy; local time is evaluated using minute-window accuracy and mean absolute error (MAE); and geographic coordinates are assessed via great-circle distance (mean, median, and thresholded ranges), following established geolocation practice (Tian et al., 2017; Vo & Hays, 2016; Arandjelovic et al., 2016). We additionally report calibration metrics (ECE and risk–coverage curves) and provide robustness stratifications, including hemisphere-flip tests and hard/OOD splits by continent, climate zone, and environment type, to reveal shortcut behaviors observed in large-scale cross-view and global datasets (Zhu et al., 2021; Astruc et al., 2024; Hou et al., 2024; Huang et al., 2024). By construction, TIMESPOT isolates capabilities that generic VLM benchmarks (e.g., CLIP (Radford et al., 2021), ViLT (Kim et al., 2021), BLIP (Li et al., 2022)) and remote-sensing–focused geospatial suites (e.g., GEOBench-VLM (Danish et al., 2024)) do not explicitly test—namely, coherent and calibrated prediction of *when* and *where* from subtle ground-level visual evidence.

## 4 EXPERIMENTS AND EVALUATION

We benchmark a diverse set of vision–language models (VLMs) to provide a comprehensive assessment of geo-temporal reasoning. The evaluated models fall into four families: *(i) proprietary* VLMs, including GPT-4o/mini, o3/o4-mini, Gemini-2/2.5-Flash, Claude 3.5 Haiku, and Mistral Medium (OpenAI, 2024; 2025; Team, 2025a; Anthropic, 2024); *(ii) open-source* VLMs spanning compact to large scales, including InternVL3, Qwen2.5-VL, Llama-3.2-Vision, Gemma-3, and GLM-4.5V (Zhu et al., 2025; Bai et al., 2025; Grattafiori et al., 2024; Gemma Team, Google DeepMind, 2025; Team et al., 2025); and *(iii) reasoning-augmented* variants that expose native *thinking* tokens while returning final structured predictions, such as o3/o4-mini, Gemini-2/2.5-Flash-Thinking, GLM-4.1V-Thinking, Kimi-VL-A3B-Thinking, and Step-3 (OpenAI, 2025; Team, 2025a; Team et al., 2025; Team, 2025b; StepFun, 2025). For open-source models, we further distinguish two parameter regimes: *small* (≤11B) and *large* (>11B).

Evaluation uses categorical accuracy for continent, country, climate zone, and environment type; local-time accuracy within ≤1 hour and mean absolute error (MAE, minutes); coordinate MAE (degrees) and mean geodesic distance (MD, km); and explicit cross-field geo-temporal consistency diagnostics. Formal metric definitions and scoring thresholds are provided in Appendix C.2.

**Table 3:** Performance of VLMs on TIMESPOT by questions. We bold and underline the best score within each model category. **Cnt.** → Continent, **Cou.** → Country, **Clim.** → Climate Zone; **Env.** → Environment Type, **Lat.°** → Latitude in degree, **Long.°** → Longitude in degree, **Dist.(km)** (MD) → mean distance from actual location in kilometers, **DLP** → Day-light phase. **Time** (Ac.) denotes accuracy, if the model predicted the time accurately within 1 hour window. **Time** (MAE) shows mean error in HH:MM format. Ac. denotes accuracy.

| Model | Geo-location Understanding | | | | | | | Temporal Understanding | | | | |
|---|---|---|---|---|---|---|---|---|---|---|---|---|
| | **Cnt.** | **Cou.** | **Clim.** | **Env.** | **Lat.°** | **Long.°** | **Dist.(km)** | **Season** | **Month** | **Time** | **Time** | **DLP** |
| | Ac.(↑) | Ac.(↑) | Ac.(↑) | Ac.(↑) | MAE (↓) | MAE (↓) | MD (↓) | Ac.(↑) | Ac.(↑) | Ac.(↑) | MAE (↓) | Ac.(↑) |
| *Proprietary Models* | | | | | | | | | | | | |
| GPT-4o-mini | 82.68 | 49.14 | 50.93 | 57.87 | 12.40 | 24.70 | 2827.07 | 47.08 | 22.34 | 30.32 | 3:54 | 31.55 |
| GPT-5-mini | 83.62 | 68.27 | **72.47** | 60.01 | 4.72 | 15.64 | 1389.79 | **58.43** | **34.27** | 21.55 | 4:10 | **44.60** |
| Gemini-2.0-Flash | 89.07 | 76.91 | 68.52 | 60.96 | 3.32 | 11.23 | 994.30 | 49.76 | 22.89 | 27.35 | 4:22 | 30.24 |
| Gemini-2.5-Flash | **90.51** | **77.25** | 71.34 | **64.32** | **3.05** | **10.38** | **917.61** | 50.92 | 23.91 | 25.15 | 3:56 | 41.92 |
| Claude 3.5 Haiku | 77.25 | 55.53 | 61.86 | 55.74 | 6.85 | 27.51 | 2269.86 | 44.12 | 19.04 | 23.09 | 4:14 | 30.93 |
| Mistral Medium 3.1 | 75.88 | 52.85 | 66.67 | 61.72 | 6.37 | 22.62 | 2045.61 | 36.84 | 15.26 | **30.73** | **3:36** | 36.01 |
| *Open-Source Models* | | | | | | | | | | | | |
| InternVL3.5-1B | 43.02 | 14.15 | 32.50 | 53.54 | 44.68 | 4378.92 | 7700.42 | 30.65 | 3.78 | 7.77 | 11:45 | 35.80 |
| InternVL3.5-2B | 60.00 | 29.41 | 51.82 | 57.80 | 13.11 | 43.71 | 3959.29 | 36.29 | 5.70 | 27.80 | 4:30 | 24.05 |
| Qwen-VL2.5-3B-Instruct | 22.40 | 13.47 | 18.83 | 44.53 | 16.18 | 130.98 | 8231.18 | 27.49 | 9.96 | 22.06 | 4:34 | 8.52 |
| InternVL3.5-4B | 60.79 | 30.12 | 57.77 | 56.74 | 15.34 | 44.15 | 4236.77 | 37.55 | 12.03 | 29.33 | 4:10 | 41.61 |
| Qwen-VL2.5-7B-Instruct | **85.70** | **73.96** | 70.86 | **75.21** | 32.94 | 21.46 | 4719.95 | **61.46** | **44.96** | 25.68 | 3:47 | **64.09** |
| Llama-3.2-11B-Vision-Instruct | 74.22 | 55.73 | 57.12 | 57.61 | 5.85 | 26.57 | 2072.35 | 43.50 | 16.68 | 25.74 | 4:18 | 43.57 |
| Gemma-3-27B-it | 79.59 | 54.02 | 60.41 | 53.12 | 6.83 | 23.58 | 2063.93 | 44.81 | 17.11 | 26.34 | 4:28 | 30.86 |
| Qwen-VL2.5-32B-Instruct | 78.56 | 57.11 | 62.95 | 60.82 | 6.27 | 24.02 | 2010.12 | 44.81 | 17.86 | **31.10** | **3:44** | 44.54 |
| Internvl3-78b | 77.46 | 53.26 | **71.61** | 61.37 | 7.42 | 23.63 | 2180.29 | 45.91 | 16.43 | 29.64 | 4:07 | 34.91 |
| Qwen-VL2.5-72B-Instruct | 77.94 | 58.28 | 65.15 | 58.14 | 5.11 | 19.33 | 1711.42 | 44.47 | 18.28 | 28.71 | 4:00 | 36.84 |
| Llama-3.2-90B-Vision-Instruct | 78.08 | 53.54 | 63.85 | 59.04 | 7.05 | 26.79 | 2284.85 | 45.15 | 19.72 | 23.33 | 4:29 | 33.88 |
| GLM-4.5V-106B-MoE | 85.32 | 69.68 | 62.09 | 62.51 | **4.23** | **14.09** | **1280.87** | 57.55 | 36.04 | 30.51 | 4:09 | 42.45 |
| *Reasoning Models* | | | | | | | | | | | | |
| o4-mini | 82.39 | 71.82 | **73.06** | 66.64 | 4.85 | 15.39 | 1359.96 | **65.81** | **48.20** | 23.91 | 4:04 | **51.79** |
| Gemini-2-Flash-Thinking | 88.66 | 76.22 | 66.73 | 59.93 | 3.44 | 11.70 | 1024.14 | 49.28 | 22.68 | 27.49 | 4:22 | 29.76 |
| Gemini-2.5-Flash-Thinking | **90.31** | **77.59** | 70.86 | 64.47 | **3.04** | **9.85** | **892.54** | 51.13 | 24.26 | 22.19 | 4:03 | 36.56 |
| Kimi-VL-A3B-Thinking-2506 | 58.90 | 40.69 | 54.84 | 59.31 | 16.00 | 39.83 | 4034.15 | 39.72 | 12.65 | 32.23 | 4:18 | 25.70 |
| GLM-4.1V-9B-Thinking | 84.44 | 68.34 | 70.19 | **68.54** | 4.34 | 23.01 | 1788.77 | 58.02 | 38.88 | **33.74** | **3:58** | 47.76 |

## 5 RESULTS AND ANALYSIS

Table 3 reports per-field accuracy grouped by model family; the best result within each family is highlighted in **bold** and underlined. While proprietary models dominate coarse spatial attributes (*continent*, *country*) and metric localization (MD), all families exhibit substantial degradation on temporal fields and frequent geo–temporal inconsistencies. Larger open-source models substantially close the gap on categorical geography, yet remain markedly weaker on *local time* and *daylight phase*. Reasoning-enhanced variants yield modest gains on *month* and *season*, suggesting better exploitation of low-salience cues, but accurate coordinate regression remains challenging under open-ended generation.

For ease of comparison, model families are color-coded in Table 3: proprietary , open-source (≤11B) , open-source (>11B) , proprietary reasoning , and open-source reasoning .

### 5.1 OVERALL PERFORMANCE

Results in Table 3 show a clear advantage for proprietary models on spatial reasoning and metric localization. Among non-*Thinking* models, **Gemini–2.5–Flash** achieves the strongest overall performance, with continent accuracy **90.51**%, country **77.25**%, climate **71.34**%, environment **64.32**%, and the lowest median distance error (MD) of **917.61** km (latitude MAE **3.05**°, longitude MAE **10.38**°). Its reasoning-enhanced variant, *Flash–Thinking*, further improves coordinate precision (latitude MAE **3.04**°, longitude MAE **9.85**°, MD **892.54** km) while preserving high country-level accuracy (**77.59**%).

Open-source models exhibit greater variance. **GLM–4.5V–106B–MoE** attains competitive country accuracy (**69.68**%) with an MD of **1280.87** km, indicating strong place recognition but weaker metric grounding. In contrast, **Qwen–VL2.5–7B–Instruct** performs well on categorical geography (continent **85.70**%, country **73.96**%, best environment accuracy **75.21**%) yet fails on coordinate estimation (latitude MAE **32.94**°, MD **4719.95** km), revealing a pronounced gap between semantic place classification and precise spatial localization.

Temporal inference remains a major bottleneck. Across all model families, time-of-day accuracy is low (typically 22–34%), with mean absolute errors of approximately four hours (3:36–4:30). Performance varies by temporal subtask: **o4–mini** achieves the highest calendar accuracy (season

Table 4: Consistency-violation and diagnostic rates across models on TIMESPOT. Lower is better.

| Model | Phase & Time (>1 h) (%) | Month & Season (%) | Season & Month (%) | Country & MD > 200 km (%) | Country & MD < 200 km (%) | Continent & Country (%) | MD > 1000 km (%) |
|---|---|---|---|---|---|---|---|
| Gpt5-Mini | 15.95 | 0.89 | 25.02 | 16.98 | 2.54 | 17.59 | 17.25 |
| intern_vl3_78B | 11.82 | 0.62 | 30.10 | 27.42 | 3.85 | 29.00 | 37.73 |
| QwenVL-3B | 0.21 | 0.82 | 18.35 | 12.78 | 0.00 | 8.93 | 95.19 |

**65.81**%, month **48.20**%), **GLM–4.1V–9B–Thinking** leads on time-of-day prediction (**33.74**%), and **Qwen–VL2.5–7B–Instruct** performs best on daylight phase (**64.09**%). These results indicate that models can often infer coarse illumination states (e.g., day vs. dusk) but struggle to recover precise local time from subtle visual cues such as solar elevation, shadow geometry, sky luminance gradients, and artificial lighting.

## 5.2 CROSS-MODEL OBSERVATIONS

Across model families, high continent- and country-level accuracy frequently coexists with large geodesic errors, suggesting reliance on coarse semantic cues (e.g., scripts, architectural styles, vegetation types) rather than true metric grounding in latitude and longitude. Reasoning-oriented models (e.g., *Gemini–2.5–Flash–Thinking*, *o4–mini*) exhibit more internally consistent coordinate and calendar predictions, indicating improved use of low-salience geo–temporal cues such as seasonal vegetation patterns and illumination regimes. Among open-source systems, increasing model scale improves categorical accuracy and reduces distance errors, yet a clear gap relative to proprietary models persists, likely reflecting differences in pretraining diversity and explicit geo–temporal supervision.

Overall, while season, month, and daylight-phase predictions reach moderate reliability, fine-grained time-of-day estimation remains weak across all models, underscoring limited exploitation of solar geometry and lighting cues. These shortcomings have direct implications for downstream applications—including autonomous navigation, situational awareness, and context-aware reasoning—where accurate, physically consistent joint inference of *where and when* is essential.

## 6 ERROR ANALYSIS

### 6.1 QUALITATIVE ERROR ANALYSIS

We summarize the main empirical regularities on TIMESPOT (1,455 images); full diagnostics appear in Appendix B. **(i) Regional vs. national)** Continent accuracy is consistently high while country accuracy drops markedly, indicating under-use of micro-geo cues; see Table 3. **(ii) Temporal precision)** Minute-level time is weak despite moderate MAE, consistent with round-time anchoring; this holds across prompt variants (Table 3). **(iii) Spatial tail risk)** Models with similar mean MD can differ sharply in MD>1000 km mass (Table 3), which governs unusably large errors. **(iv) Cross-field incoherence)** Phase–time and time–longitude mismatches persist, revealing missing soft constraints across {*phase, time, latitude*}. **(v) Field-wise cues)** Phase > time; climate/environment sit mid-band with Temperate/Urban defaults; month/season drift reflects hemispheric priors Constraint-aware joint decoding, micro-geo supervision with hard negatives, anti-anchor time regression, and hemisphere/biome-aware temporal targets are the most promising levers (details in Appendix B).

### 6.2 QUANTITATIVE ANALYSIS

Figure 3 presents representative successes and failures in geo–temporal reasoning on TIMESPOT. When salient physical cues are present—e.g., clean solar geometry in arid landscapes—the model aligns closely with ground truth (desert sunset: $|\Delta t| = 0:15$, MD = 3.5 km), indicating effective use of shadow direction and sky color for time/place inference. In contrast, scenes with low or ambiguous illumination degrade temporal accuracy: at *night*, predictions collapse to popular evening anchors (e.g., 20:30), yielding large time errors despite minor spatial drift; around *dawn/dusk*, symmetric chromatic cues trigger Sunrise↔Sunset flips (e.g., Uruguay→Argentina, $|\Delta t| \approx 11$ h), exposing weak phase disambiguation beyond hue.

Urban street views reveal a second pattern. Occlusions and canyon geometry distort shadow cues and compress apparent sun elevation, pushing hours later than reality (e.g., Turkey morning predicted as

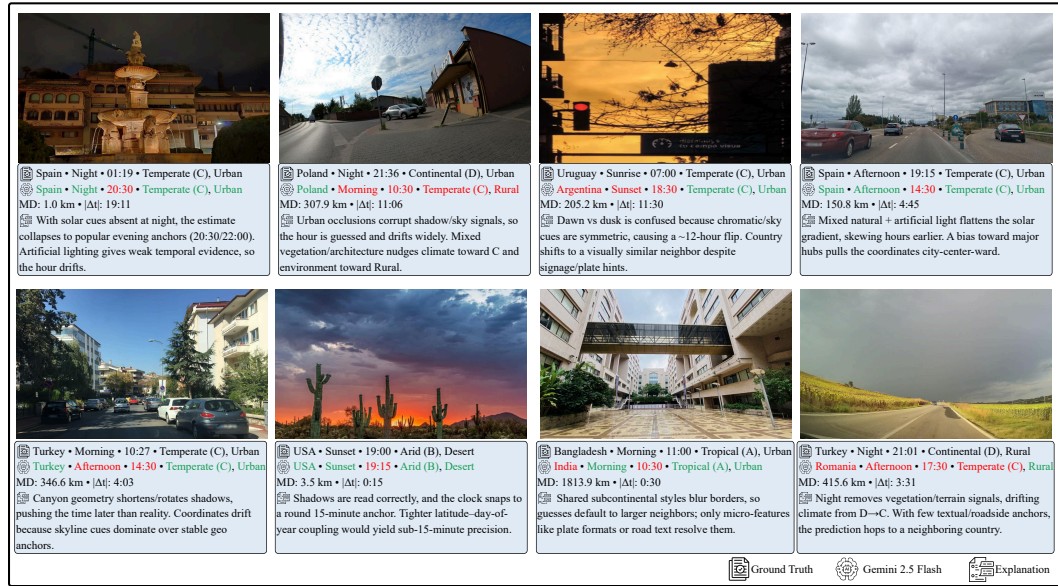

**Figure 3: Qualitative results on TIMESPOT (Gemini-2.5-Flash).** Each panel pairs the image with ground truth and model outputs—country, daylight phase, and local time—along with MD and $|\Delta t|$. Clear solar geometry yields accurate estimates (desert sunset). Night scenes and urban canyons cause round time anchoring and phase drift. Neighbor–country substitutions and limited use of coastline, topography, or micro-cues produce large MD.

afternoon; $|\Delta t| \approx 4$ h). Spatially, the model often gets the *continent* right but swaps the *country* to a regional neighbor (e.g., Bangladesh→India), suggesting reliance on broad stylistic features over micro-geocues such as signage typography, license plates, and utility hardware. We also observe environment/climate drift under low light (Poland Night predicted as Morning; Continental → Temperate), and underuse of coastal topology, where visible sea–horizon boundaries are ignored, producing large MD when predictions move inland.

Overall, these examples highlight a divide between scenes with unambiguous physical constraints (clear shadows, distinctive biomes) and those requiring integration of subtle cues (phase at twilight, fine-grained regional markers, shoreline/elevation geometry). Improving robustness likely requires (i) explicit *solar-geometry* modules that couple latitude with day-of-year, (ii) *phase-aware* temporal heads to resist round-time collapse at night, and (iii) stronger *geo-linguistic/topographic* priors (script, plate formats, coastline/elevation fingerprints) to reduce neighbor substitutions and climate drift.

# 7 CONCLUSION

Geo-temporal reasoning remains a major challenge for vision–language models in unconstrained, real-world images. TIMESPOT introduces 1,455 images across 80 countries with structured temporal (season, month, time, daylight) and geographic (continent, country, climate, environment, coordinates) fields. Evaluation reveals persistent weaknesses: even top models achieve only 77.6% country accuracy, 33.7% time-of-day accuracy, and median geodesic errors above 890 km, while weaker models fall below 50% country accuracy with extreme errors exceeding 4,700 km. Temporal predictions often conflict with solar and hemispheric constraints, and spatial predictions rely heavily on coarse priors, causing systematic neighboring-country swaps and climate misclassifications. Low-light, urban-canyon, and twilight scenes amplify failures, showing underuse of shadows, illumination gradients, and micro-geographic cues. TIMESPOT highlights the fragility of current VLMs and underscores that scaling or instruction-tuning alone is insufficient. Future work should target joint, constraint-aware reasoning, explicit solar modeling, and micro-geography supervision to improve real-world geo-temporal understanding.

ETHICS STATEMENT

TIMESPOT was created to study geo–temporal understanding in everyday, publicly observable scenes and to benchmark vision–language models on structured, verifiable outputs. Throughout curation we prioritized privacy and respectful representation. Images were sourced under licenses permitting research use or captured by the authors; items with unclear rights were excluded, and per-image license and attribution are recorded in the release. To mitigate privacy risks, we remove EXIF and other embedded metadata, blur or mask personally identifying details such as faces, license plates, and house numbers, and exclude scenes where sensitive content dominates. Geographic labels are reported at city or regional granularity, and exact dwelling locations are never included. Annotation guidelines prohibit demographic inference or stereotyping, and we audited geographic balance across continents, countries, climate zones, and environments to reduce bias. Annotators received training, quality checks, and fair compensation consistent with local norms, and could decline any image. We recognize dual-use risks in location inference; accordingly, the dataset license limits use to non-commercial research and prohibits attempts to identify individuals or private property, and we document salient failure modes to discourage deployment in safety-critical settings. To our knowledge, the work does not constitute human-subjects research under institutional policy; if required by venue or institution, we will obtain approval prior to public release. A takedown procedure and contact channel are provided so that content owners or affected parties can request removal.

REPRODUCIBILITY STATEMENT

Upon acceptance, we will ensure end-to-end reproducibility of all reported results by releasing the complete TIMESPOT image set, spatial and temporal annotations, and official train/validation/test splits with file hashes and a verification script. The code repository will provide data loaders, evaluation utilities, prompt templates, and scripts to regenerate all tables and figures directly from model outputs using a single configuration file that specifies paths, random seeds, and metric settings.

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

## A  EXTENDED RELATED WORK

**Remote sensing and aerial geospatial VLMs.**  A large body of recent work on geospatial vision–language models focuses on aerial and satellite imagery, advancing captioning, visual question answering, detection, and change analysis at planetary scale. Benchmarks such as EarthVQA (Wang et al., 2024b), RS-LLaVA (Bazi et al., 2024), RSBench/VRSBench (Li et al., 2024c), GeoChat (Kuckreja et al., 2024), RemoteCLIP (Liu et al., 2024a), RS5M/GeoRSCLIP (Zhang et al., 2024), and HRVQA (Li et al., 2024b) evaluate multi-modal reasoning and perception over remote-sensing imagery, while GEOBench-VLM (Danish et al., 2024) aggregates diverse geospatial tasks, including non-optical data and segmentation. Although these efforts provide valuable coverage for Earth observation, they primarily emphasize aerial viewpoints and do not require models to jointly reason about fine-grained temporal attributes (e.g., season, month, time, daylight phase) together with ground-level geographic context, where physical cues are weaker, noisier, and more ambiguous.

**Ground-level localization and cross-view benchmarks.**  At ground level, cross-view localization and place-recognition research has developed powerful spatial representations through retrieval and matching pipelines. Early ground-to-aerial methods (Lin et al., 2013; Workman et al., 2015b; Tian et al., 2017) and NetVLAD-style aggregation (Arandjelovic et al., 2016) laid the foundations for metric localization, followed by larger and more geographically diverse benchmarks such as CVM/Net (Hu et al., 2018; Hu & Lee, 2020). Subsequent datasets—including VIGOR (Zhu et al., 2021), OpenStreetView 5M (Astruc et al., 2024), Global Streetscapes (Hou et al., 2024), CV-Cities (Huang et al., 2024), and panoramic cross-view settings (Xia et al., 2025)—stress robustness to viewpoint, appearance, and domain gaps across continents. However, evaluation in these benchmarks is almost exclusively spatial, focusing on retrieval accuracy or coordinate error (Lin et al., 2013; Workman et al., 2015b), and does not require calibrated predictions of *when* an image was captured, nor the validation of physical constraints such as month–season–hemisphere alignment or daylight plausibility (Hu & Lee, 2020).

**Image-based geolocation with large VLMs.**  More recently, large vision–language models have significantly advanced ground-level image geolocation, but remain largely focused on *spatial inference*. LLMGeo (Wang et al., 2024d) and IMAGEO-Bench (Li et al., 2025) evaluate country-, city-, or coordinate-level localization with sophisticated prompting and structured reasoning protocols. ETHAN (Liu et al., 2024b) and subsequent agent-based evaluations (Jay et al., 2025) show that chain-of-thought reasoning and navigation tools can substantially improve spatial accuracy. FAIR-LOCATOR (Huang et al., 2025) introduces a complementary axis by revealing socio-economic and regional biases in geolocation performance. Despite their breadth and rigor, these benchmarks do not require explicit prediction of temporal attributes (e.g., month, local time, daylight phase), nor do they enforce cross-field physical consistency constraints such as month–season–hemisphere or time–daylight compatibility.

**Positioning of TIMESPOT.**  TIMESPOT is complementary to this line of work. Rather than re-evaluating whether modern VLMs can localize images spatially, it assumes strong spatial priors and instead probes whether models can *jointly infer where and when* from subtle, low-salience physical cues present in natural ground-level imagery. By requiring structured prediction over a nine-field geo–temporal schema and auditing internal consistency across geographic and temporal attributes, TIMESPOT exposes failure modes that are invisible to spatial-only benchmarks. Our results show that even state-of-the-art models with competitive geolocation accuracy exhibit large temporal errors and frequent geo–temporal inconsistencies, highlighting a critical gap in current evaluation protocols and a clear opportunity for future model and benchmark development.

## B  MORE DETAILS ON ERROR ANALYSIS

**Metrics and diagnostics.** Beyond per-field accuracy, we analyze error structure using complementary diagnostics spanning spatial, temporal, and cross-field consistency. We report categorical accuracies; local-time accuracy within $\leq 1$,hour alongside mean absolute error (MAE); mean geodesic distance (MD); and targeted failure indicators capturing physically implausible combinations, including Continent✓ & Country✗, Country✗ & MD<200,km (boundary confusions), Phase✓ & $|\Delta t| >$ 120,min (daylight–time inconsistency), and extreme spatial failures (MD>1000,km). We additionally

study correlations between time error and distance error to assess coupling between temporal and spatial inference.

**Failure modes.** Error analysis reveals several recurrent patterns. Spatial failures are often dominated by *neighboring-country substitutions* and *near-miss geography*, where predictions fall within hundreds of kilometers of the ground truth yet cross political boundaries. Heavy-tailed spatial errors emerge on non-iconic scenes lacking distinctive built or natural markers. Temporal reasoning degrades at high latitudes, where extended twilight and seasonal illumination shifts produce *phase–time anomalies*. Across models, we also observe *round-time anchoring*, with local-time predictions clustering at salient anchors (e.g., 09:00, 12:00, 18:00), indicating weak exploitation of continuous solar geometry and photometric cues.

**Ablations and remedies.** Guided by these failure modes, we evaluate several targeted interventions. *Joint, constraint-aware decoding* reduces geo–temporal inconsistencies by enforcing month–season–hemisphere and daylight–time compatibility during generation. *Micro-geo cue heads* focusing on subtle signals (e.g., license plates, curb and lane markings, shoreline typology, utility pole structure) mitigate neighboring-country confusions. *Hard-negative sampling* across borders and coast–lake lookalikes reduces near-miss spatial errors. For temporal anchoring, an *anti-anchor minutes head* combined with photometric perturbations improves fine-grained time estimation. Finally, *hemisphere- and biome-aware temporal targets* improve robustness under seasonal and latitudinal distribution shifts.

## C   MORE ON DETAILS EXPERIMENTS AND EVALUATION

### C.1   CALIBRATION

#### C.1.1   CALIBRATION METRICS

Calibration measures how closely the predicted confidence values match empirical correctness. Two metrics are used.

**Expected Calibration Error.**   For predictions with confidence scores $p_i$ and correctness indicators $\mathbf{1}(y_i = \hat{y}_i)$, the Expected Calibration Error (ECE) is

$$\text{ECE} = \sum_{m=1}^{M} \frac{|B_m|}{n} \left| \text{acc}(B_m) - \text{conf}(B_m) \right|,$$

where

$$\text{acc}(B_m) = \frac{1}{|B_m|} \sum_{i \in B_m} \mathbf{1}(y_i = \hat{y}_i), \qquad \text{conf}(B_m) = \frac{1}{|B_m|} \sum_{i \in B_m} p_i.$$

Lower ECE indicates closer alignment between confidence and accuracy.

**Risk Coverage Area Under the Curve.**   For a rejection threshold $\tau$, all predictions with confidence below $\tau$ are withheld. The resulting coverage is the fraction of retained predictions. Risk is the error rate on those retained predictions. Let

$$R(\tau), \qquad C(\tau)$$

denote risk and coverage respectively. The RC-AUC is

$$\text{RC-AUC} = \int_0^1 R(C)\, dC.$$

Lower values indicate lower risk at each coverage level.

C.1.2 CALIBRATION RESULTS

Table 5 reports ECE for all models across the four tasks. Proprietary models achieve the lowest values, and all models show larger errors for the tasks with finer granularity.

**Table 5:** Expected Calibration Error (ECE) across tasks. Lower values indicate smaller calibration error. Bold marks the best performance per task.

| Model | Continent ECE | Country ECE | Season ECE | Phase ECE |
|---|---|---|---|---|
| **Proprietary** | | | | |
| GPT 4o mini | 0.042 | 0.085 | 0.063 | 0.051 |
| Gemini 2.5 Flash | 0.021 | 0.054 | 0.041 | 0.038 |
| **Open Source** | | | | |
| Llama 3.2 90B | 0.098 | 0.142 | 0.110 | 0.095 |
| Qwen VL2.5 72B | 0.076 | 0.125 | 0.089 | 0.072 |

Table 5 show the risk coverage curves for two models. The curves steepen as task granularity increases, which indicates that confidence scores become less informative at finer resolutions. Proprietary models maintain lower risk across most coverage levels, although differences narrow in the daylight phase task, where visual cues are less distinctive.

Across both metrics, confidence reliability declines as tasks shift from continent to daylight phase classification. Although the proprietary models produce more accurate confidence estimates, all models exhibit systematic overconfidence, especially in tasks with ambiguous temporal features. These results complement the accuracy analyses in the main text by providing a detailed view of model reliability.

C.2 EVALUATION METRICS

To ensure rigorous and reproducible evaluation, we adopt metrics tailored to both geographic and temporal prediction tasks. All metrics are applied uniformly across models, and malformed outputs (e.g., missing `HH:MM` fields or unsigned coordinates) are considered incorrect, thereby preventing models from gaining undue advantage through partial responses.

**Geographic Metrics.** For categorical geographic attributes—*continent*, *country*, *climate* (Köppen–Geiger A–E) Peel et al. (2007), and *environment*—we report top-1 accuracy. For continuous localization, we measure the mean absolute error (MAE) in degrees for latitude and longitude,

$$\mathrm{MAE}_{\mathrm{lat}} = \tfrac{1}{N} \sum_{i=1}^{N} \big| \hat{\phi}_i - \phi_i \big|, \quad \mathrm{MAE}_{\mathrm{lon}} = \tfrac{1}{N} \sum_{i=1}^{N} \big| \hat{\lambda}_i - \lambda_i \big|,$$

and the mean great-circle distance (MD, km) using the haversine formula,

$$\mathrm{MD} = \tfrac{1}{N} \sum_{i=1}^{N} R \cdot 2 \arctan\!\big( \sqrt{a_i}, \sqrt{1 - a_i} \big), \ a_i = \sin^2 \tfrac{\Delta \phi_i}{2} + \cos \phi_i \cos \hat{\phi}_i \sin^2 \tfrac{\Delta \lambda_i}{2},$$

with Earth radius $R = 6371$ km and $(\phi, \lambda)$ denoting (lat, lon). These are standard geolocation metrics (Tian et al., 2017; Vo & Hays, 2016; Arandjelovic et al., 2016).

**Temporal Metrics.** For *season*, *month*, and *daylight phase* we report top-1 accuracy. For *local time* we report two complementary metrics: (i) window accuracy within $\pm 1$ hour of ground-truth,

$$\mathrm{Acc}_{\pm 1\mathrm{h}} = \tfrac{1}{N} \sum_{i=1}^{N} \mathbb{1}\!\big( |\hat{t}_i - t_i| \leq 60 \text{ min} \big),$$

and (ii) MAE in `HH:MM` after converting absolute minute errors to clock format.

# D   PROMPT TEMPLATE

---

**TimeSpot : Direct : Prompt for Answering Model**

You are a geo-spatio-temporal understanding assistant.
From the given image, answer every item below in an exact bullet list (one per line), with the format:
- field : value

Answer all fields regardless of certainty. Use one-word season and month where requested.
Fields and formats:
- season : name of the season in 1 word
- month : name of month in 1 word
- time_of_day : HH:MM (24-hour local time)
- daylight_phase : can choose one of: Sunrise, Morning, Midday, Afternoon, Sunset, Night
- continent : value
- country : value
- climate_zone : can choose one of: Tropical (A), Arid (B), Temperate (C), Continental (D), Polar (E)
- environment_type : can choose one of: Urban, Suburban, Rural, Coastal, Mountain, Desert
- coordinates_latitude : +/-DD.DDDDD (decimal degrees). DO NOT include N/S. Use +/-.
- coordinates_longitude : +/-DDD.DDDDD (decimal degrees). DO NOT include E/W. Use +/-.

Provide only the bullet list lines, nothing else.

---

**TimeSpot : Prompt for Judging Model**

You are a strict evaluator.
You will receive the ground truth and a model's answer (both in the same bullet-list format).
Compare each field. Treat abbreviations and long forms as equivalent (for example: USA == United States). Treat seasonal synonyms as equivalent (for example: Fall == Autumn). DO NOT include E/W. Use +/- in coordinates.
Return a JSON object where each field maps to a nested object:
{
  "field_name": {
    "ground_truth": "<ground truth string>",
    "model_ans": "<model answer string>",
    "evaluation": 1 or 0
  },
  ...
}
Include all fields even if the model answer is malformed. Output only valid JSON.

Ground Truth:
{json_ground_truth}

Model Answer:
{model_response_text}

**Figure 4:** Prompts used for evaluation.

# E DETAILED ANALYSIS ACROSS QUESTIONS

## E.1 DAYLIGHT PHASE ANALYSIS (SEE TABLE 6)

Table 6 reveals marked variation in model performance across different daylight phases. **Sunrise** accuracy peaks with `Gemma-3-27B` at 61.7%, whereas `Intern-VL-3-4B` (0.0%) and `Qwen-2.5-32B-Instruct` (6.4%) register the lowest scores. **Morning** is led by `Gemini-Flash-2.5-Thinking` at 65.0%, while `Kimi-VL-a3b-Thinking` falls to 7.4%. **Midday** favors `GLM-4.5 vs` with 87.9% and `Kimi-VL-a3b-Thinking` at 74.2%, yet `Gemini-Flash-2.5-Thinking` drops sharply to 3.2%. In **Afternoon**, `Qwen-2.5-32B-Instruct` leads at 70.6%, contrasted by low performance from `Kimi-VL-a3b-Thinking` (14.4%) and `Llama-3.2-90B-Vision-Instruct` (18.0%). **Sunset** peaks with `Qwen-2.5-32B-Instruct` at 55.7%. **Night** is uniformly challenging, with the highest accuracy at 34.8% (`GLM-4.5 vs` and `GPT-5-mini`) and the lowest at 20.9% (`Qwen-2.5-32B-Instruct`).

The phase difficulty profile indicates **Night** as the hardest regime for all models, with accuracies not exceeding the mid 30% range. **Sunrise** performance is volatile, spanning 0–61.7%, reflecting sensitivity to low-angle illumination and color temperature nuances. **Midday** results split models sharply: some excel with explicit shadow geometry cues, while others relying on chromatic heuristics falter.

Model specialization patterns emerge clearly. `GLM-4.5 vs` demonstrates strong solar geometry competence, dominating at **Midday** and remaining competitive at **Sunset** and **Night**. `Qwen-2.5-32B-Instruct` is **Afternoon**-focused (70.6%) and strong at **Sunset** (55.7%), but weaker at **Sunrise** and **Night**, suggesting asymmetric priors. `Gemini-Flash-2.5-Thinking` specializes in **Morning** yet struggles at **Midday**, indicating fragile temporal generalization.

Balancing overall performance, `GPT-5-mini` exhibits the most consistent profile across daylight phases, with accuracies generally in the high 40s to low 50s and mid-30s at **Night**, avoiding severe collapses. Other models show sharp peaks in certain phases (e.g., `GLM-4.5 vs` at **Midday**) but correspondingly weaker accuracy in neighboring phases.

A pronounced diurnal asymmetry is evident: `Gemini-Flash-2.5-Thinking` achieves 65.0% in **Morning** but drops to 34.8% in **Afternoon**, while `Qwen-2.5-32B-Instruct` reverses this pattern with only 20.2% in **Morning** but a strong 70.6% in **Afternoon**. This suggests reliance on hemisphere or time-of-year priors and color temperature shortcuts rather than robust reasoning about sun azimuth.

Confusability between **Sunrise** and **Sunset** is also apparent: `Qwen-2.5-32B-Instruct` and `GLM-4.5 vs` perform better at **Sunset** than **Sunrise**, consistent with warmer tone spectra being easier to interpret than pre-dawn blues. In contrast, `Gemma-3-27B` excels at **Sunrise**, indicating differing colorimetric priors across models.

Model size does not guarantee performance dominance. `Llama-3.2-90B-Vision-Instruct` exhibits oscillating scores, such as 54.8% at **Midday** versus 18.0% at **Afternoon**, underscoring the importance of instruction data and training objectives over mere parameter scale.

Two archetypes of daylight phase modeling are observed: geometry-driven models that peak sharply at **Midday** and maintain solidity at **Sunset** (e.g., `GLM-4.5 vs` and `Kimi-VL-a3b-Thinking`), and color-cue-driven models that perform better at **Morning** and **Sunset** but are brittle at **Midday** and **Night** (e.g., `Gemini-Flash-2.5-Thinking`).

The relatively low ceiling around the mid 30s percent at **Night** suggests models underutilize urban lighting, sky luminance gradients, and activity cues; in the absence of shadows, models resort to weak heuristics.

Finally, notable outliers include the 0.0% **Sunrise** accuracy of `Intern-VL-3-4B` and the 3.2% **Midday** accuracy of `Gemini-Flash-2.5-Thinking`, likely reflecting brittle mode collapses arising from data or fine-tuning skew and prompt sensitivity rather than uniformly low competence across phases.

**Table 6:** Accuracy by daylight phase (DLP) for each model on TIMESPOT. Values are percentage accuracy; blank cells indicate insufficient samples.

| Model | Sunrise | Morning | Midday | Afternoon | Sunset | Night |
|-------|---------|---------|--------|-----------|--------|-------|
| Intern-VL-3-4B | 0.00 | 12.81 | 27.42 | 65.69 | 37.62 | 28.92 |
| Llama-3.2-90B-Vision-Instruct | 23.40 | 42.36 | 54.84 | 17.98 | 46.67 | 28.57 |
| Gemini-Flash-2.5-Thinking | 25.53 | 65.02 | 3.23 | 34.76 | 48.57 | 27.53 |
| gemma_3_27B | 61.70 | 56.16 | 33.06 | 22.26 | 31.90 | 23.69 |
| glm_4.5vs | 29.79 | 36.14 | 87.90 | 35.28 | 54.76 | 34.84 |
| GPT-5-mini | 38.30 | 47.03 | 48.39 | 45.28 | 52.86 | 34.84 |
| intern_vl3_78B | 25.53 | 24.63 | 39.52 | 36.99 | 49.52 | 26.83 |
| Kimi-VL-a3b-Thinking | 21.28 | 7.39 | 74.19 | 14.38 | 45.24 | 27.18 |
| o4_mini | 14.89 | 51.72 | 31.45 | 26.71 | 49.52 | 28.57 |
| qwen_2.5_32B_instruct | 6.38 | 20.20 | 12.10 | 70.55 | 55.71 | 20.91 |

Overall, daylight phase recognition remains a nuanced task, with model behaviors shaped by distinct priors, specialized strengths, and varying robustness across illumination regimes.

E.2 SEASON ANALYSIS (SEE TABLE 7)

Table 7 reveals notable seasonal performance patterns across models. **Spring** accuracy is led by Gemma-3-27B at 44.48%, closely followed by GPT-5-mini (43.58%) and GLM-4.5 vs (42.99%). In contrast, **Summer** is dominated by GLM-4.5 vs with a commanding 84.92%, while intern vl3 78B (79.75%) and Gemini-Flash-2.5-Thinking (80.25%) also perform strongly. **Winter** peaks at GPT-5-mini (60.31%) and Llama-3.2-90B-Vision-Instruct (59.19%), whereas Qwen-2.5-32B-Instruct lags at 36.45%. Most strikingly, **Autumn** accuracy collapses to 0.00% across all models, indicating a severe and systematic failure mode.

Seasonal difficulty profiles suggest that **Summer** is the easiest season for nearly all models, with many achieving over 70% accuracy. This likely reflects strong reliance on clear cues such as high solar irradiance, saturated foliage, and short shadows. **Spring** sits in a mid-range difficulty band (approximately 19–45%), indicating moderate confusion with neighboring seasons like summer and early autumn due to transitional vegetation states and variable atmospheric conditions. **Winter** is moderately challenging but tractable, with accuracies ranging from 36–60%, potentially because of mixed phenological signals, inconsistent snow cover, and low sun angles across diverse climates. The uniquely hard **Autumn** season—where all models report 0% accuracy—points to systematic misclassification, limited or skewed label availability, or phenological overlap with adjacent seasons, highlighting bottlenecks related more to colorimetry and vegetation than solar geometry alone.

Model specialization emerges clearly: GLM-4.5 vs acts as a summer specialist, with its peak accuracy of 84.92% alongside solid winter performance (55.76%), indicating effective use of solar geometry and distinct phenological cues. Meanwhile, Gemma-3-27B leads spring accuracy at 44.48% but falls behind in summer and winter, suggesting priors better suited to transitional vegetation rather than extremes of lighting or phenology. GPT-5-mini maintains consistent top-tier performance across winter, spring, and summer, reflecting a balanced seasonal prior without catastrophic drops.

A clear trade-off is evident between models emphasizing balanced performance and those showing seasonal peaks. GPT-5-mini exhibits stable accuracy across seasons (spring, summer, winter), while GLM-4.5 vs manifests a sharp summer peak with respectable winter results. Others, such as intern vl3 4B and Kimi-VL-a3B-Thinking, show more volatile performance oscillations between seasons. Notably, most models improve markedly from spring to summer (e.g., Gemini-Flash-2.5-Thinking from 33.43% to 80.25%, intern vl3 78B from 23.58% to 79.75%), underscoring reliance on saturated green vegetation, clear skies, and sharply defined shadows in summer, whereas spring's transitional foliage and cloud variability yield weaker, noisier cues.

Winter accuracies cluster between 40–60%, led by GPT-5-mini and Llama-3.2-90B-Vision-Instruct. These results suggest partial exploitation of winter-specific cues such as low sun angles, bare trees, and snow cover where available. Failures

Table 7: Accuracy by season category for each model on TIMESPOT.

| Model | Spring | Summer | Autumn | Winter |
|---|---|---|---|---|
| Intern-VL-3-4B | 21.86 | 53.50 | 0.00 | 43.93 |
| Llama-3.2-90B-Vision-Instruct | 28.66 | 70.25 | 0.00 | 59.19 |
| Gemini-Flash-2.5-Thinking | 33.43 | 80.25 | 0.00 | 52.02 |
| gemma_3_27B | 44.48 | 48.75 | 0.00 | 43.30 |
| glm_4.5vs | 42.99 | 84.92 | 0.00 | 55.76 |
| GPT-5-mini | 43.58 | 78.95 | 0.00 | 60.31 |
| intern_vl3_78B | 23.58 | 79.75 | 0.00 | 40.81 |
| Kimi-VL-a3b-Thinking | 19.10 | 65.50 | 0.00 | 41.43 |
| o4_mini | 27.46 | 73.00 | 0.00 | 50.78 |
| qwen_2.5_32B_instruct | 36.42 | 62.50 | 0.00 | 36.45 |

likely stem from temperate and maritime climates where winter shares colorimetry with autumn and snow is absent.

The uniform failure in autumn is a red flag warranting further investigation. It may be caused by systematic confusion with summer or winter, geographically skewed or small autumn samples, or a mismatch between regional phenology and the imposed four-season taxonomy. This autumn collapse differentiates seasonal recognition from daylight phase prediction, implying that phenological and colorimetric ambiguity rather than solar angle alone forms the main bottleneck.

Model size and instruction tuning alone do not guarantee seasonal robustness. For example, `Llama-3.2-90B-Vision-Instruct` excels in winter but ranks mid-tier in spring and summer, while smaller yet well-optimized models like `GPT-5-mini` remain highly competitive. This highlights the critical role of targeted supervision and data augmentation beyond raw parameter count.

Two archetypal seasonal model behaviors emerge: phenology-driven models that perform strongly in summer and hold winter performance, exemplified by `GLM-4.5 vs`, and geometry-driven models that leverage sun angle and shadow length cues, as seen in `GPT-5-mini`'s balanced profile. Models with flatter spring performance may underutilize transitional vegetation signals.

Finally, some models display informative outlier patterns: `intern vl3 4B` struggles in spring and winter, while `Kimi-VL-a3B-Thinking` shows modest spring results, possibly reflecting sensitivity to camera pipeline effects or color casts. `Qwen-2.5-32B-Instruct` exhibits weak winter performance despite decent summer accuracy, suggesting priors biased toward warm-toned imagery and limited invariance to low irradiance conditions.

### E.3 CLIMATE ZONE ANALYSIS (SEE TABLE 8)

Table 8 reveals clear performance variation across climate zones. For the **Tropical (A)** zone, `Gemini-Flash-2.5-Thinking` leads with 86.13%, while `Kimi-VL-a3B-Thinking` is lowest at 59.85%. In the **Arid (B)** zone, `Gemini-Flash-2.5-Thinking` again performs best at 83.89%, contrasted by `Llama-3.2-90B-Vision-Instruct` at 58.33%. The **Temperate (C)** zone is topped by `o4 mini` at an impressive 90.72%, with several models clustered above 80%; `Intern-VL-3-4B` trails at 62.89%. **Continental (D)** climates prove difficult, with `intern vl3 78B` leading at 56.82% and `GPT-5-mini` close behind at 55.84%, whereas `Kimi-VL-a3B-Thinking` collapses to 2.02%. The **Polar (E)** zone remains the most challenging, with the best model, `Gemini-Flash-2.5-Thinking`, achieving only 47.83%, and `Llama-3.2-90B-Vision-Instruct` at a mere 4.35%.

The difficulty profile across zones shows that Tropical, Arid, and Temperate regions are comparatively easy, with many models exceeding 70% accuracy and a notable peak of 90.72% in Temperate. In contrast, Continental and Polar zones are consistently hard, with accuracy rarely exceeding the mid-50% range and often falling below 25%. This pattern likely reflects training and inductive biases favoring lower latitudes, higher population densities, and greener biomes.

**Table 8:** Accuracy by Köppen–Geiger climate zone (A–E) for each model on TIMESPOT.

| Model | A | B | C | D | E |
|---|---|---|---|---|---|
| Intern-VL-3-4B | 71.53 | 71.67 | 62.89 | 35.19 | 43.48 |
| Llama-3.2-90B-Vision-Instruct | 60.58 | 58.33 | 77.66 | 44.44 | 4.35 |
| Gemini-Flash-2.5-Thinking | 86.13 | 83.89 | 80.24 | 41.92 | 47.83 |
| gemma_3_27B | 78.10 | 61.67 | 83.33 | 15.40 | 34.78 |
| glm_4.5vs | 76.84 | 62.57 | 82.27 | 23.23 | 43.48 |
| GPT-5-mini | 75.91 | 75.00 | 82.47 | 55.84 | 43.48 |
| intern_vl3_78B | 76.28 | 81.67 | 78.69 | 56.82 | 13.04 |
| Kimi-VL-a3b-Thinking | 59.85 | 70.00 | 85.40 | 2.02 | 13.04 |
| o4_mini | 61.31 | 60.00 | 90.72 | 31.06 | 8.70 |
| qwen_2.5_32B_instruct | 70.44 | 74.44 | 85.91 | 21.46 | 17.39 |

The Temperate zone outlier—`o4 mini`'s 90.72% accuracy—suggests that this zone provides abundant and stable cues such as deciduous phenology, moderate sky conditions, and familiar built environments that align well with model pretraining priors.

Continental climates impose the greatest strain, exhibiting the sharpest performance spread. While two models achieve near 56%, many others fall below 25%, including a near-floor value of 2.02%. High seasonality, intermittent snow cover, and strong intra-class variance likely undermine the efficacy of single-cue heuristics like vegetation greenness or shadow length.

Polar zone results indicate pronounced brittleness: even the best model fails to surpass 50%, and several systems languish in the single digits or teens. Factors such as low sun elevations, high snow albedo, and sparse vegetation reduce discriminative texture and color signals that models typically exploit.

Strong performance in Tropical and Arid zones by `Gemini-Flash-2.5-Thinking` and `intern vl3 78B` suggests effective utilization of cues like sky clarity, aridity signatures, and architectural style. Conversely, the weak Arid zone performance of `Llama-3.2-90B-Vision-Instruct` implies limited invariance to high-contrast radiometry, dust, or haze.

Regarding balance versus specialization, `GPT-5-mini` demonstrates broad competence across zones A through D, maintaining competitiveness and avoiding catastrophic failures. In contrast, `Kimi-VL-a3B-Thinking` specializes heavily, excelling in Temperate (85.40%) but collapsing in Continental (2.02%), indicating poor transfer across diverse climates.

Model scale alone does not guarantee robustness. `Llama-3.2-90B-Vision-Instruct` ranges from strong performance in Temperate and winter-like contexts to very weak results in Polar and Arid zones, while smaller yet well-tuned models keep pace or outperform it in challenging regions.

The accuracy contrast between adjacent zones—high in Temperate and low in Continental and Polar—suggests an overreliance on mid-latitude priors. When cues deviate, such as prolonged winters, low solar zenith angles, or persistent snow cover, models tend to revert to incorrect temperate-like assumptions.

Together, the Temperate peak and Continental-Polar troughs imply current systems overfit to populous, well-photographed regions and struggle where illumination physics and surface properties diverge from those priors, highlighting critical gaps for future model development.

### E.4    ENVIRONMENT ANALYSIS (SEE TABLE 9)

Table 9 highlights notable variation in model accuracy across environment types. For the **Urban** environment, `Gemini-Flash-2.5-Thinking` leads with 75.93%, while `Gemma-3-27B` ranks lowest at 54.17%. The **Suburban** category proves broadly challenging: `GLM-4.5 vs` tops at only 46.61%, whereas `Gemini-Flash-2.5-Thinking` bottoms out at 8.47%. **Rural** scenes are dominated by `Gemma-3-27B` at 82.67%, with `Llama-3.2-90B-Vision-Instruct` trailing at 46.53%. **Coastal** accuracy peaks at 61.33% for `Qwen-2.5-32B-Instruct` but is weakest

for `Gemma-3-27B` and `Kimi-VL-a3B-Thinking` near 45.86%. In **Mountain** environments, `Llama-3.2-90B-Vision-Instruct` and `Gemini-Flash-2.5-Thinking` tie at 72.54%, while `Gemma-3-27B` lags at 38.34%. Finally, **Desert** is generally robust, with `intern vl3 78B` leading at 78.76% and `Gemma-3-27B` again trailing at 58.41%.

The difficulty profile shows **Suburban** as the most challenging environment for nearly all models, with typical accuracies between 20–40%, indicating weak priors for mixed-density built areas. By contrast, **Rural**, **Mountain**, and **Desert** are comparatively easier, with multiple models surpassing 70%. **Urban** results fall in the mid to high range (55–76%), reflecting exploitable dense man-made cues. **Coastal** scores spread moderately (46–61%), suggesting variability in coastline appearances and weather conditions.

There is a pronounced **Urban versus Suburban asymmetry**: many models perform substantially better in Urban than Suburban (e.g., `Gemini-Flash-2.5-Thinking` achieves 75.93% vs. 8.47%), implying that high-density architectural features and signage serve as stronger anchors compared to the diluted cues in mixed-density suburban neighborhoods.

**Rural** environments benefit `Gemma-3-27B` (82.67%) and show solid results for `GLM-4.5 vs` and `intern vl3 78B` (approximately 67–70%). This likely reflects the models' leveraging of vegetation structure, field patterns, and road typologies. Meanwhile, `Llama-3.2-90B-Vision-Instruct`'s weaker rural accuracy suggests sensitivity to camera pipelines or vegetation colorimetry.

The **Coastal** lead by `Qwen-2.5-32B-Instruct` (61.33%) indicates effective use of shoreline geometry, horizon–waterline relationships, and coastal infrastructure cues. Models scoring near 46–50% may underuse sky–sea radiometric contrasts and surf textures, or conflate coastal towns with generic urban–suburban scenes.

**Mountain** environment performance ties `Llama-3.2-90B-Vision-Instruct` and `Gemini-Flash-2.5-Thinking` at 72.54%, suggesting successful exploitation of high-relief silhouettes, snow lines, and atmospheric perspective. Lower-performing models likely struggle with low sun angles, haze, or mixed alpine–subalpine vegetation zones.

**Desert** robustness is evident, with leading models scoring roughly 68–79%. Strong cues such as texture monotony, dune and ripple patterns, arid sky states, and sparse vegetation are distinctive. `Gemma-3-27B`'s relative drop (58.41%) hints at sensitivity to color casts or limited exposure to arid scenes.

In terms of **balance versus specialization**, `GPT-5-mini` remains consistently mid-to-high across environments (approximately 59–66%, except Suburban at 37.61%), indicating a balanced profile. Specialists include `Gemini-Flash-2.5-Thinking` (very high Urban and Mountain but very low Suburban) and `Gemma-3-27B` (very high Rural but weak Mountain and Coastal).

The **intern family** exhibits strong Desert performance (78.76% for `intern vl3 78B`) and solid Rural/Urban scores but consistently low Suburban accuracy (22.03%). The 4B variant surprisingly remains competitive in arid-like Desert environments (77.88%), yet Suburban challenges persist.

**Informative extremes** emerge in the sharp Suburban floor (8.47%) versus Desert and Urban peaks (¿75%), revealing an over-reliance on either dense man-made cues or highly distinctive natural biomes, with failures in blended or radiometrically perturbed conditions such as Suburban or coastal/mountain environments.

**Broader Implications and Root Causes** We hypothesize that pretraining corpora over-represent iconic urban cores and visually distinctive biomes (deserts, alpine vistas) while under-representing transitional suburban morphologies. This imbalance drives sharp accuracy peaks in Urban and Desert environments but leads to Suburban collapses. Without explicit objectives that disentangle structure (e.g., built-density gradients, street hierarchy) from appearance (color and contrast), models tend to latch onto high-salience textures and signage and fail when cues mix.

The relative success in Rural settings suggests some models effectively leverage vegetation and road-layout priors, though sensitivity to color pipeline variations (white balance, HDR) lowers robustness across different cameras and seasons.

**Table 9:** Accuracy by environment type for each model on TimeSpot.

| Model | Urban | Suburban | Rural | Coastal | Mountain | Desert |
|---|---|---|---|---|---|---|
| Intern-VL-3-4B | 60.65 | 27.35 | 54.46 | 46.96 | 60.62 | 77.88 |
| Llama-3.2-90B-Vision-Instruct | 54.94 | 27.97 | 46.53 | 55.80 | 72.54 | 76.99 |
| Gemini-Flash-2.5-Thinking | 75.93 | 8.47 | 62.87 | 49.17 | 72.54 | 70.80 |
| gemma_3_27B | 54.17 | 27.12 | 82.67 | 45.86 | 38.34 | 58.41 |
| glm_4.5vs | 64.19 | 46.61 | 69.65 | 57.46 | 58.03 | 72.57 |
| GPT-5-mini | 62.50 | 37.61 | 55.94 | 58.89 | 65.80 | 68.14 |
| intern_vl3_78B | 66.67 | 22.03 | 66.83 | 54.14 | 58.55 | 78.76 |
| Kimi-VL-a3b-Thinking | 68.83 | 20.34 | 57.43 | 45.86 | 59.07 | 70.80 |
| o4_mini | 63.27 | 32.20 | 66.83 | 50.28 | 54.40 | 70.80 |
| qwen_2.5_32B_instruct | 63.73 | 26.27 | 62.38 | 61.33 | 61.66 | 75.22 |

Coastal performance variance likely stems from under-modeled sky–sea interactions, horizon geometry, and weather/season coupling; introducing auxiliary prediction heads for horizon detection, sky state, and water presence could improve robustness.

Mountain performance benefits from silhouette and relief cues but degrades under haze, low solar elevation, or snow cover perturbations, suggesting photometric augmentation tailored to altitude conditions may be beneficial.

Finally, Suburban environments require structure-aware supervision (e.g., density gradients, land-use mixtures) and region-aware conditioning to reduce misclassification into Urban or Rural categories. Multi-task consistency linking environment type with daylight phase, climate zone, and season can regularize predictions in ambiguous scenes without eroding performance in clearly distinctive environments.

E.5    Asia Country Analysis (see Table 10)

Table 10 reveals significant variation in model performance across Asian countries. Notable country leaders include Singapore and Bangladesh, each achieving perfect 100.0% accuracy with `Gemini-Flash-2.5-Thinking` and `GLM-4.5 vs`, respectively. Other top performers are Japan (92.54%), Turkey (90.91%), and India (89.66%), all led by `Gemini-Flash-2.5-Thinking`. In contrast, multiple countries exhibit near-zero minima in certain models, such as `Intern-VL-3-4B` in Saudi Arabia and Sri Lanka, `Kimi-VL-a3B-Thinking` in Bangladesh, and `Gemma-3-27B` in Kyrgyzstan.

Regarding **model wins by country**, `Gemini-Flash-2.5-Thinking` dominates Japan, India, South Korea, Turkey, Philippines (tied with `GPT-5-mini`), Sri Lanka, Mongolia, and Singapore. `GLM-4.5 vs` leads in Russia, China, Thailand, Bangladesh, Kyrgyzstan, and Myanmar. `GPT-5-mini` takes Kazakhstan and ties in Saudi Arabia and the Philippines, while `Intern-VL-3-78B` leads in Nepal, and `Llama-3.2-90B-Vision-Instruct` leads in Bhutan.

The **regional patterning** shows East and parts of Southeast Asia (Japan, South Korea, China; Singapore, Philippines, Thailand) generally achieve high accuracy across models. Central and high-altitude countries (Nepal, Bhutan, Kyrgyzstan, Mongolia) show greater variance and model-specific collapses. West and Central Asia present mid-range accuracy with occasional peaks, such as Saudi Arabia in the mid-60s for `Gemini-Flash-2.5-Thinking` and `GPT-5-mini`, and Kazakhstan favoring `GPT-5-mini`.

**Balanced versus spiky profiles** are evident: `GPT-5-mini` maintains stable mid-to-high performance across many countries (e.g., Russia 72.06%, China 75.86%, India 74.14%, South Korea 85.00%, Kazakhstan 66.67%), avoiding catastrophic lows. In contrast, `Kimi-VL-a3B-Thinking` and `Intern-VL-3-4B` show spiky outcomes—strong in some regions but near-zero in others—highlighting sensitivity to scene distribution or prompt formatting.

**Large-scale model size is not decisive**: `Llama-3.2-90B-Vision-Instruct` posts strong results in Japan and Bhutan but remains mid-tier or weak elsewhere (e.g., Kazakhstan 20.00%, Bangladesh 22.22%), indicating that parameter count alone does not ensure country-level robustness.

Countries with distinctive urban or coastal features—such as Japan, Singapore, and the Philippines—tend to yield higher accuracy, reflecting a **urbanized and coastal advantage**. Models appear to leverage stable architectural styles, traffic infrastructure, coastline geometry, and signage conventions that are more uniform and thus easier to learn.

Conversely, **terrain and inland variance** is pronounced in mountainous or landlocked countries (Nepal, Bhutan, Kyrgyzstan, Mongolia), where illumination variability, snowline effects, mixed vegetation bands, and fewer salient man-made landmarks lead to larger accuracy spreads across models.

`Gemini-Flash-2.5-Thinking` is a consistent leader in populous and coastal contexts, remaining competitive inland. Meanwhile, `GLM-4.5 vs` excels in continental and mixed terrain settings (Russia, China, Thailand, Bangladesh, Kyrgyzstan, Myanmar). These complementary wins suggest different cue priors—color and structure cues for `Gemini-Flash-2.5-Thinking` versus geometry and terrain-texture cues for `GLM-4.5 vs`.

Instances of ties (e.g., Saudi Arabia: `Gemini-Flash-2.5-Thinking` and `GPT-5-mini` both at 64.71%; Philippines: same models tied at 88.89%) indicate shared strengths on specific country-level cues. However, large intra-country spreads—such as Bangladesh ranging from 0.00 to 100.0%—expose brittleness related to camera pipelines, lighting conditions, or sub-region diversity.

Countries featuring mid-latitude or mixed biomes (e.g., Iran, Turkey, Mongolia) display moderate average accuracy but wide per-model dispersion, implying models overfit to subsets of cues (seasonal colorimetry, skyline geometry) and falter when these shift within the country.

**Broader Implications and Root Causes**   We hypothesize that country-level accuracy peaks are driven by pretraining data density and visual homogeneity: countries with abundant, consistent urban–coastal imagery yield higher scores, while terrain-diverse or sparsely documented countries induce greater variance. Without explicit objectives modeling sun geometry, terrain context, and region-aware semantics, models rely heavily on color and texture correlates that vary across sub-regions and seasons within a single country.

Instruction tuning appears to emphasize textual plausibility over physically grounded cues, resulting in performance collapses in countries where colorimetry or scene composition deviates from high-data priors. Variations in camera pipeline parameters (white balance, HDR) and weather regimes (haze, monsoon clouds, snow) further perturb radiometric signals, explaining extreme lows.

To mitigate these issues, we recommend: (i) region-balanced data curation within each country, (ii) photometric and phenological augmentations tailored to local climate regimes, (iii) auxiliary prediction heads for horizon detection, sun elevation, and terrain class to disentangle illumination from land cover, and (iv) consistency constraints that couple country identity with climate zone, season, and daylight phase. These steps should reduce intra-country variance while preserving strengths in highly photographed locales.

E.6   EUROPE COUNTRY ANALYSIS (SEE TABLE 11)

Table 11 reveals marked variation in model performance across European countries, with several microstates exhibiting ceiling effects—many models achieve near-perfect accuracy (e.g., Ireland, Switzerland, Netherlands; Monaco and Malta for select models). Conversely, notable dips occur for `o4 mini` in the United Kingdom (13.33%) and multiple zeros for `Intern-VL-3-4B` in small countries such as Denmark, Finland, and Luxembourg.

In **Western Europe**, France, Italy, Spain, and the United Kingdom demonstrate consistently strong results for `Gemini-Flash-2.5-Thinking` (ranging 80–97%) and often for `GPT-5-mini` (60–100%). Meanwhile, `Llama-3.2-90B-Vision-Instruct` performs reliably in the UK (90%) and Italy (60%) but shows greater variability elsewhere (e.g., France 42%).

**Central Europe** displays more dispersion: Germany is a high-accuracy hub with `Gemini-Flash-2.5-Thinking` at 93.33%, `Intern-VL-3-78B` and `Qwen` at 86.67%, and

**Table 10:** Country-level accuracy (%) by model for all available countries in Asia.

| Country | Intern-VL-3-4B | Llama-3.2-90B-Vision-Instruct | gemini_flash 2.5_thinking | gemma_3_27B | glm_4.5vs | GPT-5-mini | intern_vl3_78B | kimi_vl_a3b thinking | o4_mini | qwen_2.5_32B instruct |
|---|---|---|---|---|---|---|---|---|---|---|
| Russia | 52.17 | 52.17 | 73.91 | 55.07 | 78.26 | 72.06 | 76.81 | 40.58 | 59.42 | 68.12 |
| Japan | 61.19 | 77.61 | 92.54 | 77.61 | 88.06 | 86.36 | 83.58 | 70.15 | 76.12 | 77.61 |
| China | 55.17 | 51.72 | 81.03 | 58.62 | 84.48 | 75.86 | 81.03 | 55.17 | 68.97 | 67.24 |
| India | 31.03 | 67.24 | 89.66 | 70.69 | 71.93 | 74.14 | 72.41 | 41.38 | 72.41 | 75.86 |
| South Korea | 25.00 | 75.00 | 87.50 | 67.50 | 67.50 | 85.00 | 62.50 | 60.00 | 65.00 | 62.50 |
| Turkey | 36.36 | 45.45 | 90.91 | 66.67 | 69.70 | 69.70 | 57.58 | 57.58 | 45.45 | 54.55 |
| Philippines | 14.81 | 48.15 | 88.89 | 74.07 | 59.26 | 88.89 | 33.33 | 44.44 | 74.07 | 51.85 |
| Iran | 5.26 | 45.00 | 70.00 | 35.00 | 50.00 | 65.00 | 50.00 | 25.00 | 50.00 | 35.00 |
| Myanmar | 5.56 | 5.56 | 22.22 | 33.33 | 55.56 | 27.78 | 16.67 | 11.11 | 16.67 | 11.11 |
| Bhutan | 5.88 | 64.71 | 58.82 | 47.06 | 52.94 | 47.06 | 35.29 | 29.41 | 35.29 | 52.94 |
| Saudi Arabia | 0.00 | 47.06 | 64.71 | 29.41 | 62.50 | 64.71 | 41.18 | 29.41 | 29.41 | 29.41 |
| Sri Lanka | 0.00 | 29.41 | 76.47 | 23.53 | 70.59 | 52.94 | 17.65 | 5.88 | 17.65 | 29.41 |
| Nepal | 18.75 | 31.25 | 68.75 | 56.25 | 62.50 | 50.00 | 75.00 | 56.25 | 56.25 | 56.25 |
| Kazakhstan | 0.00 | 20.00 | 60.00 | 40.00 | 60.00 | 66.67 | 6.67 | 13.33 | 40.00 | 13.33 |
| Mongolia | 23.08 | 61.54 | 76.92 | 46.15 | 46.15 | 69.23 | 53.85 | 15.38 | 53.85 | 61.54 |
| Thailand | 38.46 | 46.15 | 53.85 | 46.15 | 84.62 | 53.85 | 61.54 | 53.85 | 46.15 | 46.15 |
| Singapore | 20.00 | 80.00 | 100.00 | 90.00 | 90.00 | 90.00 | 70.00 | 20.00 | 90.00 | 80.00 |
| Bangladesh | 11.11 | 22.22 | 66.67 | 55.56 | 100.00 | 55.56 | 11.11 | 0.00 | 33.33 | 11.11 |
| Kyrgyzstan | 0.00 | 22.22 | 44.44 | 0.00 | 44.44 | 33.33 | 33.33 | 11.11 | 11.11 | 22.22 |

Gemma-3-27B at 75.56%. Poland and Czechia exhibit broader spreads, with mid-to-high wins for Gemini-Flash-2.5-Thinking (84% in Poland) but mixed outcomes for other models, reflecting sensitivity to local appearance diversity.

In the **Nordics and Atlantic edge**, Finland, Norway, Iceland, and Ireland reach very high plateaus, frequently hitting 100% for Gemini-Flash-2.5-Thinking and GPT-5-mini. However, smaller models like Intern-VL-3-4B show volatility, with 0% scores in several Nordic cases, suggesting sample size limitations and sensitivity to winter lighting conditions.

**Balkans and microstates** reveal distinct patterns: Croatia achieves triple 100% (for GLM-4.5 vs, GPT-5-mini, and o4 mini), while North Macedonia remains uniformly low, and Slovenia centers around a single leader, Gemini-Flash-2.5-Thinking (83.33%). Small states such as Andorra, Vatican City, and Luxembourg show sparse, highly discrete performance bands tied to limited sample sizes.

Regarding **coastal versus inland cues**, countries with strong coastal identities (Spain, Portugal, Norway) tend to yield high marks for models exploiting horizon geometry, maritime infrastructure, and coastal skylines (Gemini-Flash-2.5-Thinking, GPT-5-mini). Inland nations with mixed terrain (Hungary, Slovakia, Lithuania) show more fragmentation and lower median accuracies.

**Recurrent leaders** include Gemini-Flash-2.5-Thinking, frequently topping major economies (UK 96.67%, Germany 93.33%, Spain 90%) and several small states (Estonia, Monaco at 100%). GPT-5-mini repeatedly achieves 100% in northern and microstate settings (Iceland, Ireland, Netherlands, Switzerland), indicative of robust priors for well-photographed European scenes.

Concerning **model stability versus spikes**, GPT-5-mini exhibits steady high performance with few catastrophic failures, whereas o4 mini and Intern_vl3 variants display spikes and collapses depending on the country (e.g., UK 13.33% for o4 mini versus 100% in Croatia), consistent with sensitivity to particular cue distributions.

An **alphabet and signage bias** is evident: countries with uniform Latin typography and standardized road signage (Netherlands, Ireland, UK) yield higher scores, suggesting that text and iconography serve as strong anchors. Failures in low-sample countries likely arise when such cues are occluded or absent.

Russia stands out with **mid-to-high accuracy** for geometry-leaning models (GLM-4.5 vs 75.00%) and balanced models (GPT-5-mini 67.86%), but wide inter-model spread reflects challenges posed by broad biome variety and seasonal changes within the country.

**Context and Likely Drivers** These European results reflect a confluence of data-density effects and cue availability. Countries with abundant iconic urban imagery and standardized visual systems (signage, architecture, transit) favor models tuned to stable mid-latitude priors. Small states and low-sample splits yield quantized results, with a few successes at 100% and misses at 0%. High-latitude lighting regimes (long twilight, low solar elevation) can either aid prediction through predictable skylight geometry or hinder it due to photometric drift, depending on a model's radiometric invariance.

Over-reliance on textual artifacts likely benefits countries such as the UK, Ireland, and the Netherlands but falters when signage is atypical or occluded. Large parameter counts do not guarantee uniform wins; rather, training data composition and augmentation breadth appear more decisive.

To improve robustness, we recommend per-country balance checks to avoid microstate skew, photometric normalization for high-latitude lighting, and auxiliary prediction heads coupling terrain, horizon, and sky state with country labels. Finally, consistency constraints linking country identity with climate zone, season, and daylight phase should curb implausible combinations and stabilize performance in heterogeneous interiors such as the Balkans and Central/Eastern Europe.

**Table 11:** Country-level accuracy (%) by model for all available countries in Europe (selected).

| Country | Intern-VL-3-4B | Llama-3.2-90B-Vision-Instruct | gemini_flash 2.5_thinking | gemma_3_27B | glm_4.5vs | GPT-5-mini | intern_vl3_78B | kimi_vl_a3b thinking | o4_mini | qwen_2.5_32B instruct |
|---|---|---|---|---|---|---|---|---|---|---|
| Italy | 24.62 | 60.00 | 81.54 | 58.46 | 69.23 | 72.31 | 46.15 | 50.77 | 64.62 | 56.92 |
| France | 40.00 | 42.00 | 80.00 | 42.00 | 60.00 | 60.00 | 46.00 | 62.00 | 42.00 | 50.00 |
| Germany | 33.33 | 57.78 | 93.33 | 75.56 | 75.56 | 55.56 | 86.67 | 53.33 | 66.67 | 86.67 |
| United Kingdom | 60.00 | 90.00 | 96.67 | 36.67 | 83.33 | 93.33 | 90.00 | 80.00 | 13.33 | 93.33 |
| Russia | 53.57 | 35.71 | 67.86 | 35.71 | 75.00 | 67.86 | 71.43 | 46.43 | 64.29 | 53.57 |
| Poland | 20.00 | 48.00 | 84.00 | 64.00 | 64.00 | 60.00 | 36.00 | 24.00 | 36.00 | 64.00 |
| Spain | 45.00 | 50.00 | 90.00 | 60.00 | 70.00 | 65.00 | 80.00 | 45.00 | 40.00 | 90.00 |
| Vatican City | 0.00 | 40.00 | 80.00 | 0.00 | 40.00 | 50.00 | 0.00 | 30.00 | 10.00 | 40.00 |
| Andorra | 0.00 | 50.00 | 83.33 | 16.67 | 83.33 | 66.67 | 0.00 | 0.00 | 0.00 | 16.67 |
| Estonia | 0.00 | 33.33 | 100.00 | 66.67 | 16.67 | 33.33 | 16.67 | 33.33 | 0.00 | 33.33 |
| Iceland | 0.00 | 66.67 | 83.33 | 83.33 | 83.33 | 100.00 | 83.33 | 66.67 | 66.67 | 83.33 |
| Luxembourg | 0.00 | 66.67 | 50.00 | 16.67 | 33.33 | 50.00 | 0.00 | 0.00 | 0.00 | 33.33 |
| Malta | 0.00 | 50.00 | 83.33 | 66.67 | 100.00 | 66.67 | 33.33 | 16.67 | 83.33 | 83.33 |
| Monaco | 0.00 | 66.67 | 100.00 | 100.00 | 83.33 | 66.67 | 16.67 | 0.00 | 66.67 | 66.67 |
| North Macedonia | 0.00 | 0.00 | 33.33 | 0.00 | 33.33 | 16.67 | 0.00 | 0.00 | 16.67 | 0.00 |
| Slovenia | 0.00 | 33.33 | 83.33 | 33.33 | 33.33 | 16.67 | 16.67 | 16.67 | 33.33 | 16.67 |
| Croatia | 0.00 | 50.00 | 75.00 | 50.00 | 100.00 | 100.00 | 50.00 | 50.00 | 100.00 | 50.00 |
| Denmark | 0.00 | 0.00 | 100.00 | 50.00 | 50.00 | 75.00 | 25.00 | 75.00 | 75.00 | 75.00 |
| Ireland | 0.00 | 100.00 | 100.00 | 100.00 | 100.00 | 100.00 | 100.00 | 100.00 | 75.00 | 100.00 |
| Lithuania | 0.00 | 25.00 | 50.00 | 0.00 | 25.00 | 25.00 | 25.00 | 0.00 | 0.00 | 25.00 |
| Slovakia | 0.00 | 25.00 | 25.00 | 0.00 | 50.00 | 0.00 | 0.00 | 0.00 | 25.00 | 25.00 |
| Belgium | 0.00 | 33.33 | 33.33 | 0.00 | 33.33 | 66.67 | 0.00 | 33.33 | 0.00 | 0.00 |
| Czechia | 0.00 | 33.33 | 33.33 | 66.67 | 66.67 | 33.33 | 33.33 | 33.33 | 0.00 | 33.33 |
| Finland | 0.00 | 100.00 | 100.00 | 66.67 | 66.67 | 100.00 | 0.00 | 0.00 | 33.33 | 66.67 |
| Greece | 33.33 | 0.00 | 66.67 | 66.67 | 33.33 | 33.33 | 0.00 | 0.00 | 33.33 | 0.00 |
| Hungary | 0.00 | 0.00 | 66.67 | 33.33 | 33.33 | 33.33 | 33.33 | 33.33 | 33.33 | 0.00 |
| Norway | 66.67 | 33.33 | 100.00 | 66.67 | 66.67 | 100.00 | 66.67 | 33.33 | 33.33 | 100.00 |
| Portugal | 33.33 | 33.33 | 66.67 | 66.67 | 33.33 | 66.67 | 33.33 | 66.67 | 33.33 | 33.33 |
| Romania | 0.00 | 33.33 | 100.00 | 0.00 | 100.00 | 66.67 | 0.00 | 0.00 | 33.33 | 0.00 |
| Sweden | 0.00 | 66.67 | 66.67 | 33.33 | 33.33 | 100.00 | 33.33 | 0.00 | 33.33 | 33.33 |
| Netherlands | 50.00 | 100.00 | 100.00 | 100.00 | 50.00 | 100.00 | 100.00 | 50.00 | 100.00 | 100.00 |
| Serbia | 0.00 | 50.00 | 50.00 | 0.00 | 50.00 | 100.00 | 50.00 | 0.00 | 50.00 | 50.00 |
| Switzerland | 100.00 | 100.00 | 100.00 | 100.00 | 100.00 | 100.00 | 100.00 | 50.00 | 100.00 | 100.00 |
| Turkey | 0.00 | 50.00 | 50.00 | 50.00 | 100.00 | 100.00 | 50.00 | 0.00 | 50.00 | 50.00 |

### E.7 NORTH AMERICA COUNTRY ANALYSIS (SEE TABLE 12)

Table 12 highlights the USA as a high-accuracy anchor across nearly all models, with scores approximately ranging from 85% to 94%; `Gemini-Flash-2.5-Thinking` leads at 93.88%. Canada forms a distinct second tier, with accuracies between roughly 54% and 82%, while Mexico proves notably tougher, with best performances at 56.67% (`Gemini-Flash-2.5-Thinking`) and 51.72% (`GLM-4.5 vs`).

Island nations exhibit contrasting patterns. Cuba attains multiple perfect or near-perfect scores (e.g., `Llama-3.2-90B`, `Gemini-Flash-2.5-Thinking`, `GLM-4.5 vs`, `GPT-5-mini`, `o4 mini`), likely reflecting either distinctive, easy-to-key cues or small-$N$ quantization effects. The Dominican Republic presents a split profile: `Gemini-Flash-2.5-Thinking` achieves 90%, and `GPT-5-mini` 70%, whereas several other models perform poorly, indicating model-specific sensitivity to scene distribution.

Within Mesoamerica, Guatemala reveals divergent priors: `GPT-5-mini` (90%) and `GLM-4.5 vs` (80%) outperform `Gemini-Flash-2.5-Thinking` (60%), while other models linger at roughly 20–30%. Panama clusters mid-range overall (most models scoring 50–80%), but `Intern-VL-3-78B` notably drops to 10%, highlighting a significant failure mode.

Puerto Rico occupies a tight mid-band (approximately 40–60%) across models, implying the presence of useful yet non-dominant coastal or urban cues; no single model dominates this regime.

Regarding model leadership by country, `Gemini-Flash-2.5-Thinking` leads the USA and ties or wins in several island settings. `GLM-4.5 vs` consistently performs well in Canada (76%)

and reaches 100% in Cuba. `GPT-5-mini` records standout wins in Guatemala (90%) and remains competitive in both the USA and Canada.

In terms of stability, `GPT-5-mini` emerges as the steadiest performer with few catastrophic failures and mostly upper-mid scores. By contrast, `Intern-VL-3-78B` exhibits wide volatility (e.g., 87.76% in the USA versus 0–10% in the Dominican Republic and Panama), and the 4B variant frequently records low scores (Mexico 3.33%, several zeros elsewhere).

High scores in the USA and Cuba likely reflect rich coastal–urban signatures, standardized signage, and consistent road furniture. In contrast, Mexico and parts of Central America present greater intra-country heterogeneity (lighting regimes, street textures, informal signage), challenging models that rely heavily on uniform text or skyline patterns.

Model size alone does not dictate leaderboard positions: `Llama-3.2-90B` excels in the USA and Cuba but is mid-pack or weaker in Canada and Mexico. Smaller, well-tuned systems (`GPT-5-mini`, `GLM-4.5 vs`) often keep pace or outperform larger counterparts in several countries.

Outlier performances, such as `o4 mini`'s strong USA score (85.71%) but weaker results (20–50%) in smaller countries, suggest that models lean on high-data priors and revert to chance-level accuracy where those cues diminish.

**Practical Takeaway** For deployment across North America, an ensemble approach weighting `Gemini-Flash-2.5-Thinking` in the USA and islands, `GLM-4.5 vs` in Canada, and `GPT-5-mini` in Central America would effectively hedge against single-model brittleness.

**Interpretation and Likely Drivers** These patterns reflect a mix of data-density effects and cue stability. The USA's consistently high accuracy aligns with abundant, homogeneous training imagery and standardized visual systems (signage, lane markings, storefront styles). Canada's slightly lower scores likely arise from latitude-driven illumination changes and snow/overcast regimes that perturb color priors; models with greater radiometric robustness (`GLM-4.5 vs`, `GPT-5-mini`) gain advantage.

Mexico and parts of Central America introduce varied street morphologies and photometric conditions (haze, intense sun, mixed materials), reducing the reliability of text and skyline heuristics. Island peaks such as Cuba may reflect both small-$N$ discretization and distinctive coastal silhouettes and maritime infrastructure that certain models exploit.

Large performance swings in `intern_vl3` variants point to prompt-format or finetuning mismatches under domain shift. Mitigations include country-balanced sampling within regions, photometric augmentations tailored for high-irradiance and overcast extremes, auxiliary heads for horizon and sky state to disentangle lighting from semantics, and lightweight ensembling guided by per-country validation to avoid collapse on single-prior dependencies.

**Table 12:** Country-level accuracy (%) by model for all available countries in North America.

| Country | Intern-VL-3-4B | Llama-3.2-90B-Vision-Instruct | gemini_flash 2.5_thinking | gemma_3_27B | glm_4.5vs | GPT-5-mini | intern_vl3_78B | kimi_vl_a3b thinking | o4_mini | qwen_2.5_32B instruct |
|---|---|---|---|---|---|---|---|---|---|---|
| USA | 68.37 | 92.35 | 93.88 | 88.27 | 88.78 | 89.80 | 87.76 | 69.39 | 85.71 | 87.24 |
| Canada | 16.00 | 54.00 | 82.00 | 40.00 | 76.00 | 64.00 | 54.00 | 22.00 | 72.00 | 64.00 |
| Mexico | 3.33 | 46.67 | 56.67 | 43.33 | 51.72 | 46.67 | 33.33 | 26.67 | 43.33 | 26.67 |
| Cuba | 30.00 | 100.00 | 100.00 | 90.00 | 100.00 | 100.00 | 80.00 | 40.00 | 100.00 | 70.00 |
| Dominican Republic | 20.00 | 60.00 | 90.00 | 20.00 | 20.00 | 70.00 | 0.00 | 10.00 | 50.00 | 10.00 |
| Guatemala | 0.00 | 30.00 | 60.00 | 30.00 | 80.00 | 90.00 | 20.00 | 20.00 | 20.00 | 30.00 |
| Panama | 0.00 | 40.00 | 80.00 | 70.00 | 70.00 | 70.00 | 10.00 | 30.00 | 50.00 | 60.00 |
| Puerto Rico | 0.00 | 40.00 | 60.00 | 50.00 | 60.00 | 50.00 | 30.00 | 40.00 | 30.00 | 50.00 |

E.8 SOUTH AMERICA COUNTRY ANALYSIS (SEE TABLE 13)

Table 13 reveals high scores in Chile and Brazil, moderate performance bands in Argentina, Colombia, and Peru, and clear difficulty in Bolivia and Uruguay. Chile peaks with `Gemini-Flash-2.5-Thinking` at 90%, followed by `Llama-3.2-90B` at 80%, and `o4 mini` and `GPT-5-mini` at 75%. Brazil is co-led by `Gemini-Flash-2.5-Thinking` and `GLM-4.5 vs` at 90%, while Ecuador performs strongly with `Gemini-Flash-2.5-Thinking` (85%) and `GLM-4.5 vs` (84.21%).

In contrast, Bolivia exhibits uniformly low results, with the best model, `GPT-5-mini`, reaching only 30%, and many models scoring near zero. Uruguay also proves challenging, topping at just 40% for `GLM-4.5 vs` and `GPT-5-mini`. Argentina and Peru sit in a mid-tier range, led by `Gemini-Flash-2.5-Thinking` at 80% and 75% respectively, with `GLM-4.5 vs` trailing at 62.5% and 65%. Colombia shows a balanced high performance, favoring `GLM-4.5 vs` at 76.67%, closely followed by `Gemini-Flash-2.5-Thinking` (66.67%) and `GPT-5-mini` (60%).

Across countries, `Gemini-Flash-2.5-Thinking` emerges as the most frequent winner, with `GLM-4.5 vs` often placing first or second, especially in Colombia, Brazil, and Ecuador. Although `GPT-5-mini` rarely leads outright, it consistently posts stable upper-mid results and uniquely tops the low-signal Bolivia and ties for Uruguay. Some models such as `Intern-VL-3-4B` and `Kimi-VL-a3B-Thinking` show repeated near-zero scores in Argentina, Ecuador, Brazil, and Uruguay, while `intern_vl3_78B` oscillates widely—from 65% in Chile to near zero in Argentina and Uruguay. Notably, `Llama-3.2-90B` performs excellently in Chile (80%) and solidly elsewhere but drops to 10% in Uruguay, indicating that parameter count alone does not explain leaderboard positions.

These patterns reflect several key factors. High performance in Chile and Brazil aligns with strong coastal skylines, standardized road furniture, and dense urban signatures—visual cues that many models reliably exploit. Conversely, Central Andean scenes in Bolivia and parts of Peru and Ecuador involve challenging high-elevation lighting conditions, snowlines, and thin atmosphere radiometry, where models tuned primarily to mid-latitude, sea-level appearances underperform. Furthermore, heterogeneity within country labels—such as mixed street morphologies, informal signage, and diverse building materials—reduces the effectiveness of text and skyline heuristics, mirroring patterns seen in Mexico and other regions.

Robustness differences emerge across models: `GPT-5-mini`'s steadiness suggests better invariance to camera pipelines and illumination shifts, while other models with sharp performance peaks appear to rely on brittle chromatic or texture priors. The training data composition also influences outcomes; pretraining often centers on well-photographed major coastal cities, biasing models toward those cues and limiting performance in rural, highland, or low-contrast areas. The presence of near-zero and sudden jumps in scores, particularly in smaller sample sizes like Uruguay, underscores the importance of reporting variance alongside accuracy, ideally with per-country sample sizes and confidence intervals.

To address these challenges, climate- and altitude-aware augmentation strategies (e.g., modeling low sun angles, haze, high albedo, and desaturated palettes), explicit training objectives for horizon and sun geometry, and region-conditioned adapters could reduce failures in Andean interiors and heterogeneous regions. Practically, a simple ensemble approach—weighting `Gemini-Flash-2.5-Thinking` for Brazil and Chile, `GLM-4.5 vs` for Colombia and Ecuador, and `GPT-5-mini` as a stabilizer for Bolivia and Uruguay—would hedge against single-model brittleness while maintaining efficiency.

**Reflection.** These findings emphasize the critical importance of environmental and geographic diversity for model robustness. The pronounced differences between visually consistent coastal urban centers and challenging high-altitude rural interiors reveal significant limitations in current pretraining distributions, which disproportionately favor well-photographed, stable scenes. The volatility observed in less specialized models highlights the need for architectural and training strategies that incorporate regional characteristics explicitly. Addressing these gaps with more balanced datasets, climate- and altitude-aware augmentations, and modular model components will be crucial for achieving geographic generalization and equitable performance across South America and similar diverse regions.

# F    EXAMPLES OF TIMESPOT DATASET

**Table 13:** Country-level accuracy (%) by model for all available countries in South America.

| Country | Intern-VL-3-4B | Llama-3.2-90B-Vision-Instruct | gemini_flash 2.5_thinking | gemma_3_27B | glm_4.5vs | GPT-5-mini | intern_vl3_78B | kimi_vl_a3b thinking | o4_mini | qwen_2.5_32B instruct |
|---|---|---|---|---|---|---|---|---|---|---|
| Argentina | 0.0 | 40.00 | 80.00 | 35.0 | 62.50 | 55.0 | 2.50 | 12.5 | 22.50 | 37.50 |
| Colombia | 20.0 | 46.67 | 66.67 | 40.0 | 76.67 | 60.0 | 23.33 | 20.0 | 43.33 | 43.33 |
| Chile | 20.0 | 80.00 | 90.00 | 55.0 | 70.00 | 75.0 | 65.00 | 15.0 | 75.00 | 65.00 |
| Ecuador | 0.0 | 50.00 | 85.00 | 35.0 | 84.21 | 70.0 | 20.00 | 10.0 | 30.00 | 40.00 |
| Peru | 25.0 | 60.00 | 75.00 | 55.0 | 65.00 | 60.0 | 45.00 | 50.0 | 45.00 | 55.00 |
| Bolivia | 0.0 | 20.00 | 20.00 | 0.0 | 20.00 | 30.0 | 0.00 | 0.0 | 0.00 | 10.00 |
| Brazil | 0.0 | 60.00 | 90.00 | 20.0 | 90.00 | 60.0 | 50.00 | 20.0 | 50.00 | 50.00 |
| Uruguay | 0.0 | 10.00 | 30.00 | 10.0 | 40.00 | 40.0 | 0.00 | 0.0 | 0.00 | 0.00 |

**Desert Sunset USA**

**Picture**

**Ground Truth**

- Season: Summer,
- Month: June,
- Time_of_day: 19:00:00,
- Daylight_phase: Sunset,
- Continent: North America,
- Country: USA,
- Climate_zone: Arid (B),
- Environment_type: Desert,
- Coordinates_latitude: 32.2217,
- Coordinates_longitude: -110.9265

**GPT-5-mini**

- Season: Summer,
- Month: July,
- Time_of_day: 19:15,
- Daylight_phase: Sunset,
- Continent: North America,
- Country: United States,
- Climate_zone: Arid (B),
- Environment_type: Desert,
- Coordinates_latitude: +32.25000,
- Coordinates_longitude: -111.00000

**Intern-VL3.5-2B**

- Season: Winter
- Month: December
- Time_of_day: 17:30
- Daylight_phase: Sunset
- Continent: North America
- Country: United States (Arizona)
- Climate_zone: Arid (B)
- Environment_type: Desert
- Coordinates_latitude: +33.421
- Coordinates_longitude: -117.726

**Intern-VL2.5-72B**

- Season: Summer,
- Month: June,
- Time_of_day: 19:00,
- Daylight_phase: Sunset,
- Continent: North America,
- Country: United States,
- Climate_zone: Arid (B),
- Environment_type: Desert,
- Coordinates_latitude: +33.4484,
- Coordinates_longitude: -111.9385

**Figure 5:** Example of TIMESPOT dataset — Desert Sunset USA.

## Mountain Town Morning In India

### Picture

### Ground Truth

- Season: Summer
- Month: June
- Time_of_day: 09:00:00
- Daylight_phase: Morning
- Continent: Asia
- Country: India
- Climate_zone: Arid (B)
- Environment_type: Rural
- Coordinates_latitude: 34.15258
- Coordinates_longitude: 77.57705

### GPT-5-mini

- Season: Summer
- Month: August
- Time_of_day: 17:30
- Daylight_phase: Afternoon
- Continent: Asia
- Country: India
- Climate_zone: Arid (B)
- Environment_type: Mountain
- Coordinates_latitude: +34.15260
- Coordinates_longitude: +077.57700

### Intern-VL3.5-2B

- Season: Spring
- Month: July
- Time_of_day: 12:00
- Daylight_phase: Midday
- Continent: Asia
- Country: Pakistan
- Climate_zone: Arid (B)
- Environment_type: Mountain
- Coordinates_latitude: +34.835
- Coordinates_longitude: +74.679

### Intern-VL2.5-72B

- Season: Autumn
- Month: October
- Time_of_day: 16:00
- Daylight_phase: Afternoon
- Continent: Asia
- Country: India
- Climate_zone: Arid (B)
- Environment_type: Mountain
- Coordinates_latitude: +34.19999
- Coordinates_longitude: +77.58333

**Figure 6:** Example of TIMESPOT dataset — Mountain Town Morning In India.

## Urban Morning Street Scene In Turkey

### Picture

### Ground Truth

- Season: Fall,
- Month: September,
- Time_of_day: 10:27,
- Daylight_phase: Morning,
- Continent: Asia,
- Country: Turkey,
- Climate_zone: Temperate (C),
- Environment_type: Urban,
- Coordinates_latitude: 39.896579640955,
- Coordinates_longitude: 32.860911162601

### GPT-5-mini

- Season: Autumn,
- Month: October,
- Time_of_day: 11:30,
- Daylight_phase: Morning,
- Continent: Europe,
- Country: Turkey,
- Climate_zone: Temperate (C),
- Environment_type: Urban,
- Coordinates_latitude: +39.93340,
- Coordinates_longitude: +32.85970

### Intern-VL3.5-2B

- Season: Winter
- Month: October
- Time_of_day: 12:00
- Daylight_phase: Midday
- Continent: Europe
- Country: Turkey
- Climate_zone: Temperate
- Environment_type: Urban
- Coordinates_latitude: 41.167
- Coordinates_longitude: 28.917

### Intern-VL2.5-72B

- Season: Summer,
- Month: July,
- Time_of_day: 14:00,
- Daylight_phase: Afternoon,
- Continent: Europe,
- Country: Turkey,
- Climate_zone: Temperate (C),
- Environment_type: Urban,
- Coordinates_latitude: +41.02370,
- Coordinates_longitude: +28.98390

**Figure 7:** Example of TIMESPOT dataset — Urban Morning Street Scene In Turkey.

## Urban Facade In Thailand Afternoon

**Picture**

**Ground Truth**

- Season: Summer,
- Month: August,
- Time_of_day: 20:00:00,
- Daylight_phase: Night,
- Continent: Asia,
- Country: Thailand,
- Climate_zone: Tropical (A),
- Environment_type: Urban,
- Coordinates_latitude: 12.242756,
- Coordinates_longitude: 102.5174734

**GPT-5-mini**

- Season: Autumn,
- Month: November,
- Time_of_day: 19:30,
- Daylight_phase: Night,
- Continent: Asia,
- Country: China,
- Climate_zone: Temperate (C),
- Environment_type: Urban,
- Coordinates_latitude: +31.23041,
- Coordinates_longitude: +121.47370

**Intern-VL3.5-2B**

- Season: Winter
- Month: December
- Time_of_day: 23:00
- Daylight_phase: Night
- Continent: Asia
- Country: China
- Climate_zone: Temperate (C)
- Environment_type: Urban
- Coordinates_latitude: 36.89000
- Coordinates_longitude: 117.33100

**Intern-VL2.5-72B**

- Season: Autumn,
- Month: October,
- Time_of_day: 20:00,
- Daylight_phase: Night,
- Continent: Asia,
- Country: China,
- Climate_zone: Temperate (C),
- Environment_type: Urban,
- Coordinates_latitude: +31.2304,
- Coordinates_longitude: +121.4737

**Figure 8:** Example of TIMESPOT dataset — Urban Facade In Thailand Afternoon.

## Snowy Urban Night In USA

**Picture**

**Ground Truth**

- Season: Winter,
- Month: February,
- Time_of_day: 22:00:00,
- Daylight_phase: Night,
- Continent: North America,
- Country: USA,
- Climate_zone: Continental (D),
- Environment_type: Urban,
- Coordinates_latitude: 39.7392,
- Coordinates_longitude: -104.9903

**GPT-5-mini**

- Season: Winter,
- Month: January,
- Time_of_day: 20:30,
- Daylight_phase: Night,
- Continent: North America,
- Country: United States,
- Climate_zone: Continental (D),
- Environment_type: Urban,
- Coordinates_latitude: +41.87811,
- Coordinates_longitude: -087.62980

**Intern-VL3.5-2B**

- Season: Winter
- Month: January
- Time_of_day: 18:00
- Daylight_phase: Night
- Continent: North America
- Country: United States
- Climate_zone: Temperate (C)
- Environment_type: Urban
- Coordinates_latitude: -34.50167
- Coordinates_longitude: -73.85638

**Intern-VL2.5-72B**

- Season: Winter,
- Month: December,
- Time_of_day: 20:00,
- Daylight_phase: Night,
- Continent: North America,
- Country: United States,
- Climate_zone: Continental (D),
- Environment_type: Urban,
- Coordinates_latitude: +41.8781,
- Coordinates_longitude: -87.6298

**Figure 9:** Example of TIMESPOT dataset — Snowy Urban Night In USA.

## Sunny Roadside Afternoon In Turkey

**Picture**

**Ground Truth**

- Season: Summer,
- Month: July,
- Time_of_day: 15:39,
- Daylight_phase: Afternoon,
- Continent: Asia,
- Country: Turkey,
- Climate_zone: Temperate (C),
- Environment_type: Urban,
- Coordinates_latitude: 40.189871,
- Coordinates_longitude: 25.895506

**GPT-5-mini**

- Season: Summer,
- Month: July,
- Time_of_day: 13:30,
- Daylight_phase: Midday,
- Continent: Asia,
- Country: Turkey,
- Climate_zone: Temperate (C),
- Environment_type: Rural,
- Coordinates_latitude: +38.35000,
- Coordinates_longitude: +27.20000

**Intern-VL3.5-2B**

- Season: Spring
- Month: April
- Time_of_day: 12:00
- Daylight_phase: Midday
- Continent: Asia
- Country: Israel
- Climate_zone: Temperate (C)
- Environment_type: Rural
- Coordinates_latitude: 31.854
- Coordinates_longitude: 34.915

**Intern-VL2.5-72B**

- Season: Summer,
- Month: July,
- Time_of_day: 12:00,
- Daylight_phase: Midday,
- Continent: Europe,
- Country: Greece,
- Climate_zone: Temperate (C),
- Environment_type: Rural,
- Coordinates_latitude: +37.95000,
- Coordinates_longitude: +23.71667

**Figure 10:** Example of TIMESPOT dataset — Sunny Roadside Afternoon In Turkey.

## Rural River Afternoon In Russia

**Picture**

**Ground Truth**

- Season: Summer,
- Month: August,
- Time_of_day: 13:00:00,
- Daylight_phase: Afternoon,
- Continent: Asia,
- Country: Russia,
- Climate_zone: Continental (D),
- Environment_type: Rural,
- Coordinates_latitude: 56.793902,
- Coordinates_longitude: 118.273021

**GPT-5-mini**

- Season: Summer,
- Month: July,
- Time_of_day: 14:30,
- Daylight_phase: Afternoon,
- Continent: Asia,
- Country: Russia,
- Climate_zone: Continental (D),
- Environment_type: Rural,
- Coordinates_latitude: +60.50000,
- Coordinates_longitude: +101.50000

**Intern-VL3.5-2B**

- Season: Spring
- Month: May
- Time_of_day: 14:00
- Daylight_phase: Morning
- Continent: Europe
- Country: Ukraine
- Climate_zone: Continental (D)
- Environment_type: Mountain
- Coordinates_latitude: 38.4
- Coordinates_longitude: -116.6

**Intern-VL2.5-72B**

- Season: Summer,
- Month: June,
- Time_of_day: 12:00,
- Daylight_phase: Midday,
- Continent: Europe,
- Country: Russia,
- Climate_zone: Continental (D),
- Environment_type: Rural,
- Coordinates_latitude: +55.00000,
- Coordinates_longitude: +100.00000

**Figure 11:** Example of TIMESPOT dataset — Rural River Afternoon In Russia.

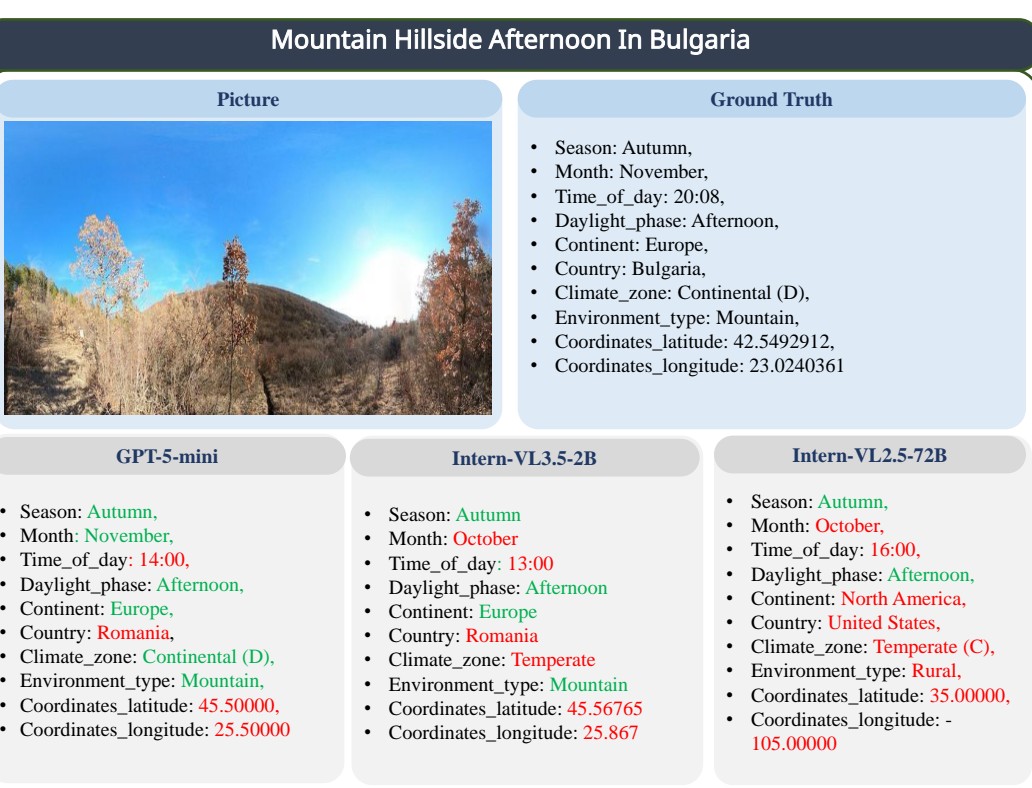

**Figure 12:** Example of TIMESPOT dataset — Urban Riverside Night In Ecuador.

**Figure 13:** Example of TIMESPOT dataset — Mountain Hillside Afternoon In Bulgaria.

## Cliffside Lake Afternoon In China

### Picture

### Ground Truth

- Season: Spring,
- Month: March,
- Time_of_day: 15:00:00,
- Daylight_phase: Afternoon,
- Continent: Asia,
- Country: China,
- Climate_zone: Continental (D),
- Environment_type: Suburban,
- Coordinates_latitude: 34.85714,
- Coordinates_longitude: 118.07065

### GPT-5-mini

- Season: Autumn,
- Month: November,
- Time_of_day: 15:30,
- Daylight_phase: Afternoon,
- Continent: Asia,
- Country: China,
- Climate_zone: Continental (D),
- Environment_type: Mountain,
- Coordinates_latitude: +37.00000,
- Coordinates_longitude: +112.50000

### Intern-VL3.5-2B

- Season: Autumn
- Month: September
- Time_of_day: 12:00
- Daylight_phase: Midday
- Continent: Asia
- Country: Iran
- Climate_zone: Arid (B)
- Environment_type: Mountain
- Coordinates_latitude: +31.0
- Coordinates_longitude: +48.9

### Intern-VL2.5-72B

- Season: Autumn,
- Month: October,
- Time_of_day: 10:00,
- Daylight_phase: Morning,
- Continent: Asia,
- Country: China,
- Climate_zone: Continental (D),
- Environment_type: Mountain,
- Coordinates_latitude: +35.00000,
- Coordinates_longitude: +110.00000

**Figure 14:** Example of TIMESPOT dataset — Cliffside Lake Afternoon In China.

## Forest Tram Midday In Russia

### Picture

### Ground Truth

- Season: Summer,
- Month: July,
- Time_of_day: 12:39,
- Daylight_phase: Midday,
- Continent: Asia,
- Country: Russia,
- Climate_zone: Continental (D),
- Environment_type: Rural
- Coordinates_latitude: 51.841950850008,
- Coordinates_longitude: 107.61454671992

### GPT-5-mini

- Season: Summer,
- Month: June,
- Time_of_day: 17:30,
- Daylight_phase: Afternoon,
- Continent: Europe,
- Country: Russia,
- Climate_zone: Continental (D),
- Environment_type: Rural,
- Coordinates_latitude: +55.75000,
- Coordinates_longitude: +37.60000

### Intern-VL3.5-2B

- Season: Summer
- Month: July
- Time_of_day: 10:00
- Daylight_phase: Morning
- Continent: Europe
- Country: Poland
- Climate_zone: Temperate (C)
- Environment_type: Rural
- Coordinates_latitude: +52.418
- Coordinates_longitude: +21.007

### Intern-VL2.5-72B

- Season: Summer,
- Month: June,
- Time_of_day: 12:00,
- Daylight_phase: Midday,
- Continent: Europe,
- Country: Russia,
- Climate_zone: Continental (D),
- Environment_type: Rural,
- Coordinates_latitude: +55.75583,
- Coordinates_longitude: +37.61730

**Figure 15:** Example of TIMESPOT dataset — Forest Tram Midday In Russia.

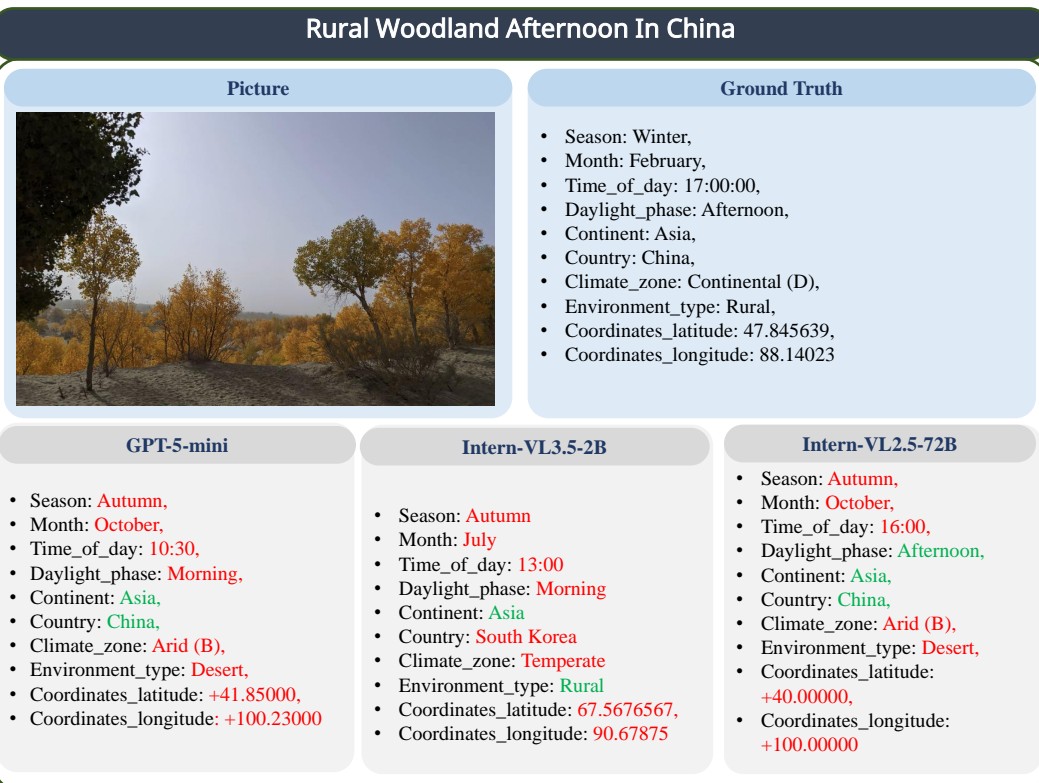

**Figure 16:** Example of TIMESPOT dataset — Coastal Walkway Afternoon In Italy.

**Figure 17:** Example of TIMESPOT dataset — Rural Woodland Afternoon In China.

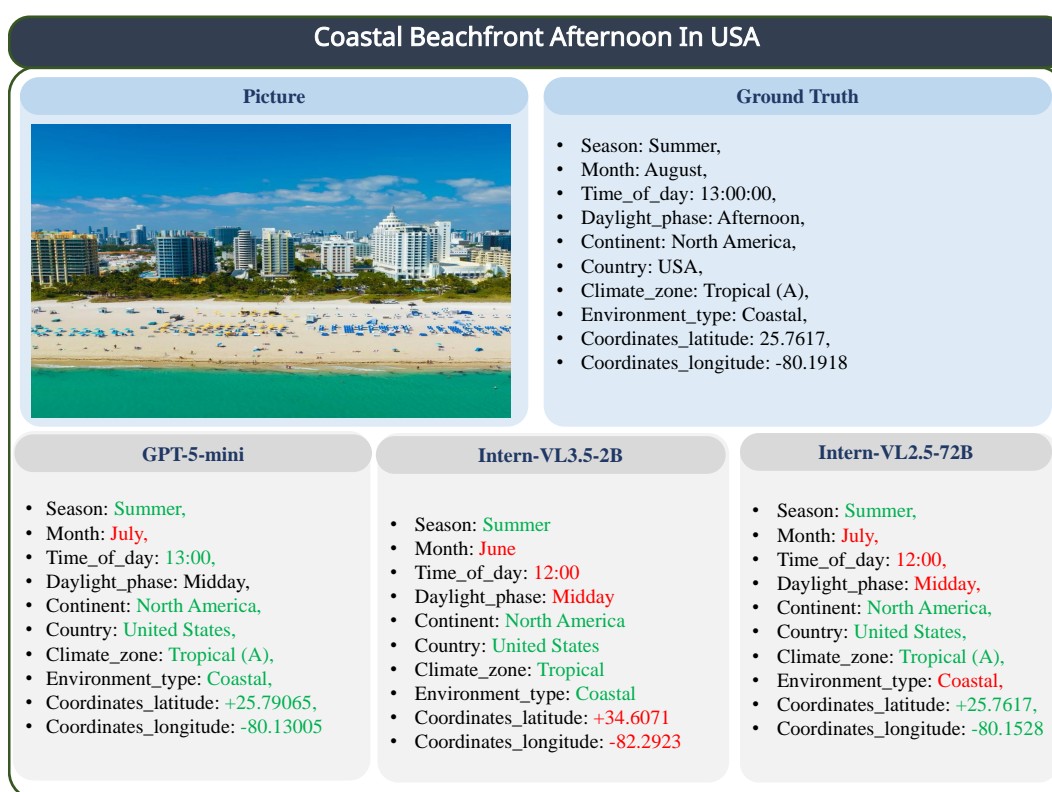

**Figure 18:** Example of TIMESPOT dataset — Coastal Beachfront Afternoon In USA.

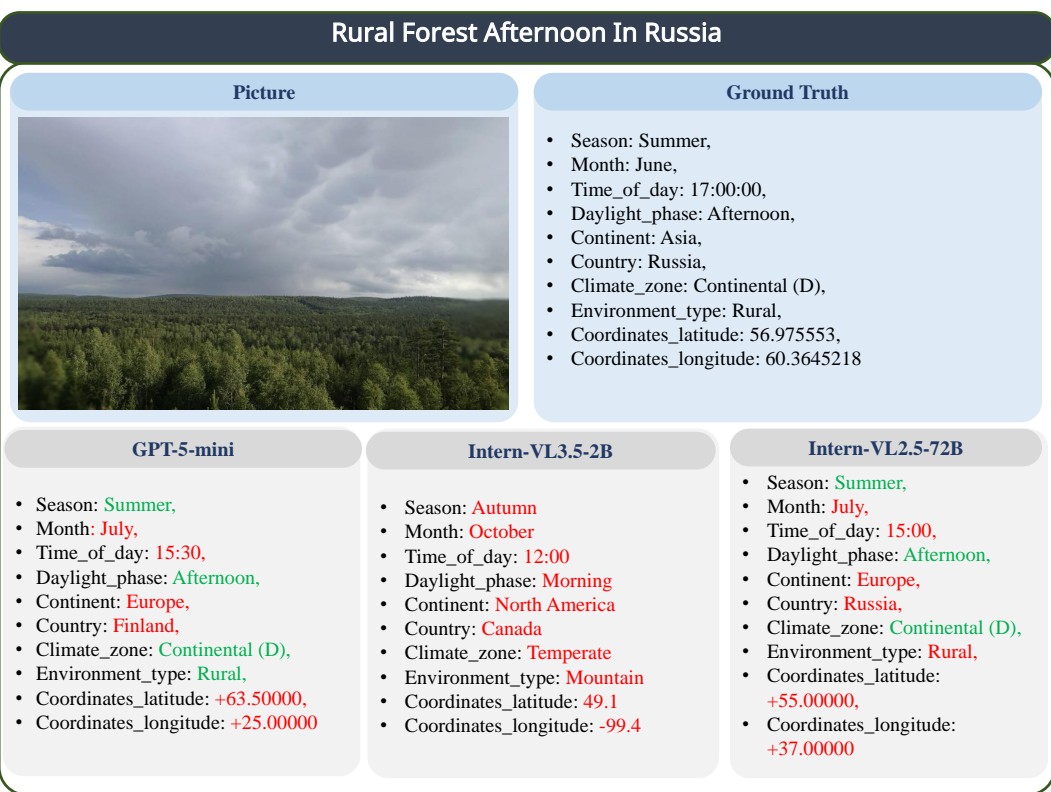

**Figure 19:** Example of TIMESPOT dataset — Rural Forest Afternoon In Russia.

## Rural Hillside Midday In Nepal

### Picture

### Ground Truth

- Season: Fall,
- Month: October,
- Time_of_day: 12:00:00,
- Daylight_phase: Midday,
- Continent: Asia,
- Country: Nepal,
- Climate_zone: Temperate (C),
- Environment_type: Rural,
- Coordinates_latitude: 28.5388196,
- Coordinates_longitude: 82.777235

### GPT-5-mini

- Season: Autumn,
- Month: October,
- Time_of_day: 12:30,
- Daylight_phase: Midday,
- Continent: Asia,
- Country: Nepal,
- Climate_zone: Temperate (C),
- Environment_type: Mountain,
- Coordinates_latitude: +28.70000,
- Coordinates_longitude: +84.00000

### Intern-VL3.5-2B

- Season: Spring
- Month: May
- Time_of_day: 11:00
- Daylight_phase: Morning
- Continent: Asia
- Country: Nepal
- Climate_zone: Arid (B)
- Environment_type: Rural
- Coordinates_latitude: +29.545
- Coordinates_longitude: +81.805

### Intern-VL2.5-72B

- Season: Autumn,
- Month: October,
- Time_of_day: 14:00,
- Daylight_phase: Afternoon,
- Continent: Asia,
- Country: Nepal,
- Climate_zone: Temperate (C),
- Environment_type: Rural,
- Coordinates_latitude: +28.12500,
- Coordinates_longitude: +83.66667

**Figure 20:** Example of TIMESPOT dataset — Rural Hillside Midday In Nepal.

## Mountain Monastery Morning In Bhutan

### Picture

### Ground Truth

- Season: Summer,
- Month: May,
- Time_of_day: 07:00:00,
- Daylight_phase: Morning,
- Continent: Asia,
- Country: Bhutan,
- Climate_zone: Temperate (C),
- Environment_type: Mountain,
- Coordinates_latitude: 27.4888,
- Coordinates_longitude: 89.3587

### GPT-5-mini

- Season: Spring,
- Month: May,
- Time_of_day: 09:30,
- Daylight_phase: Morning,
- Continent: Asia,
- Country: Bhutan,
- Climate_zone: Temperate (C),
- Environment_type: Mountain,
- Coordinates_latitude: +27.49139,
- Coordinates_longitude: +89.36394

### Intern-VL3.5-2B

- Season: Winter
- Month: January
- Time_of_day: 12:00
- Daylight_phase: Midday
- Continent: Asia
- Country: Nepal
- Climate_zone: Temperate (C)
- Environment_type: Mountain
- Coordinates_latitude: 27.7675
- Coordinates_longitude: 88.283

### Intern-VL2.5-72B

- Season: Spring,
- Month: April,
- Time_of_day: 10:00,
- Daylight_phase: Morning,
- Continent: Asia,
- Country: Bhutan,
- Climate_zone: Temperate (C),
- Environment_type: Mountain,
- Coordinates_latitude: +27.10510,
- Coordinates_longitude: +90.48820

**Figure 21:** Example of TIMESPOT dataset — Mountain Monastery Morning In Bhutan.

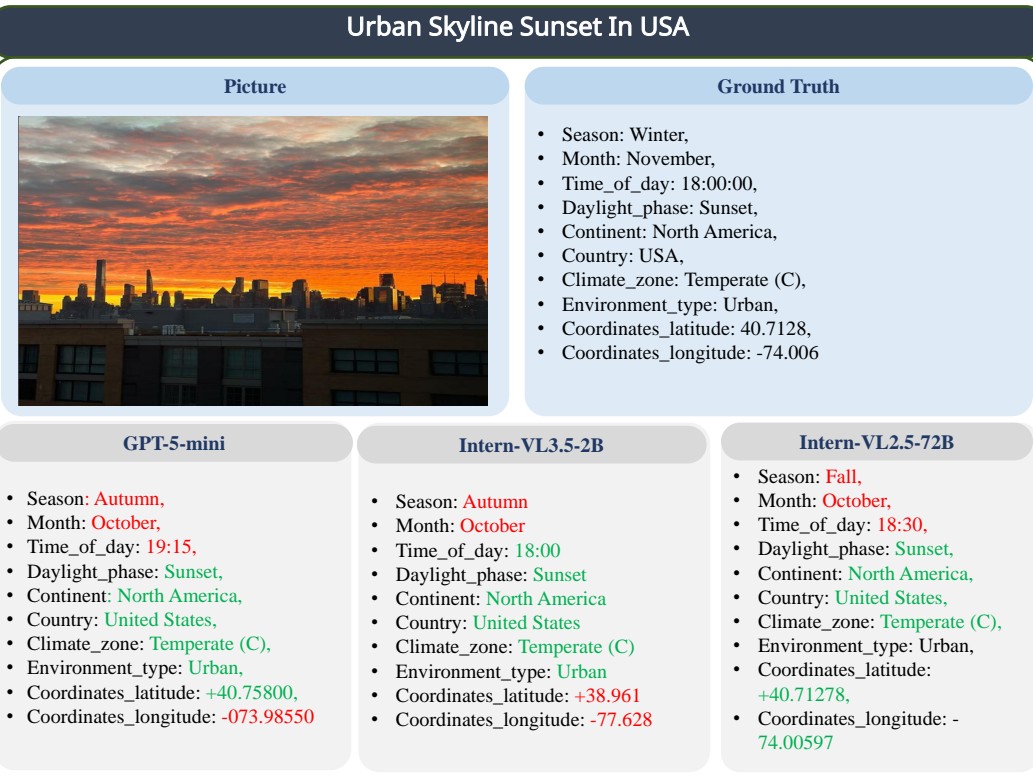

**Figure 22:** Example of TIMESPOT dataset — Urban Plaza Sunset In Germany.

**Figure 23:** Example of TIMESPOT dataset — Urban Skyline Sunset In USA.

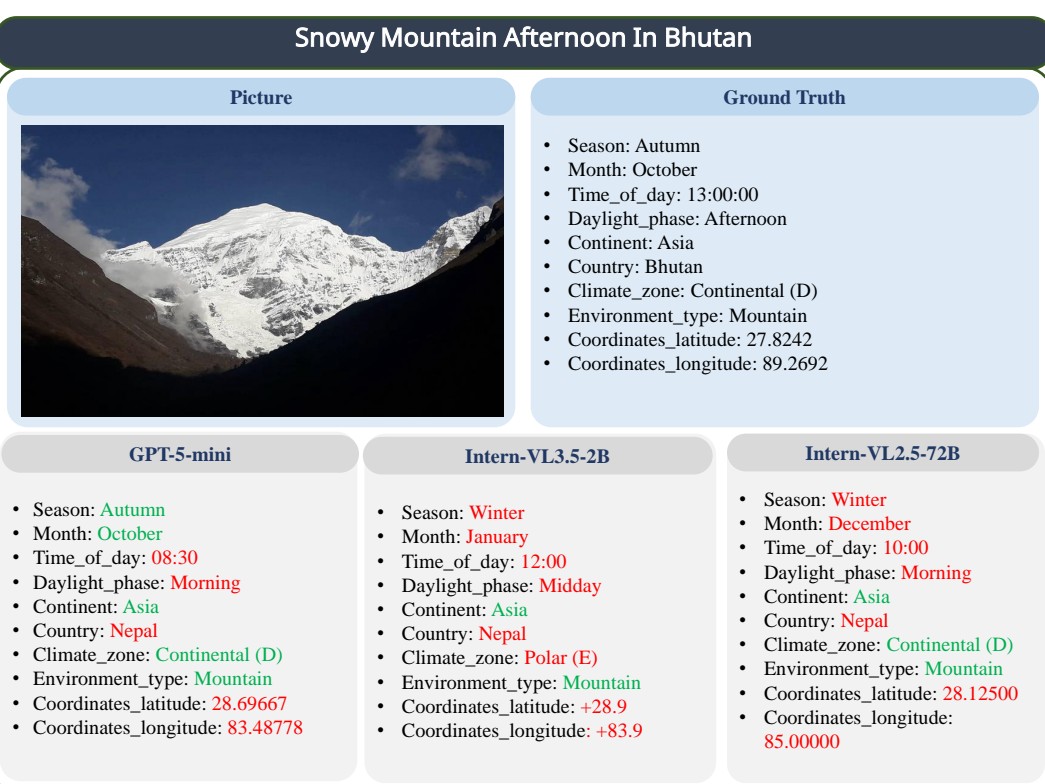

**Figure 24:** Example of TIMESPOT dataset — Forest Road Afternoon In France.

**Figure 25:** Example of TIMESPOT dataset — Snowy Mountain Afternoon In Bhutan.

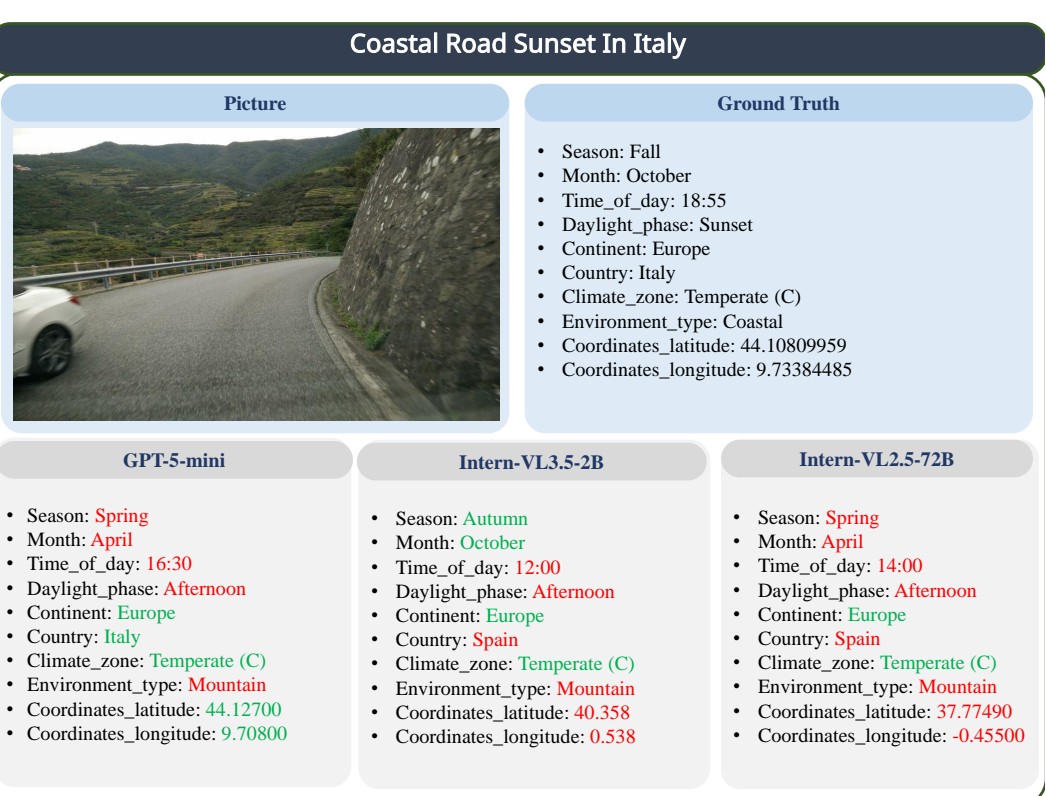

**Figure 26:** Example of TIMESPOT dataset — Coastal Beach Afternoon In Philippines.

**Figure 27:** Example of TIMESPOT dataset — Coastal Road Sunset In Italy.

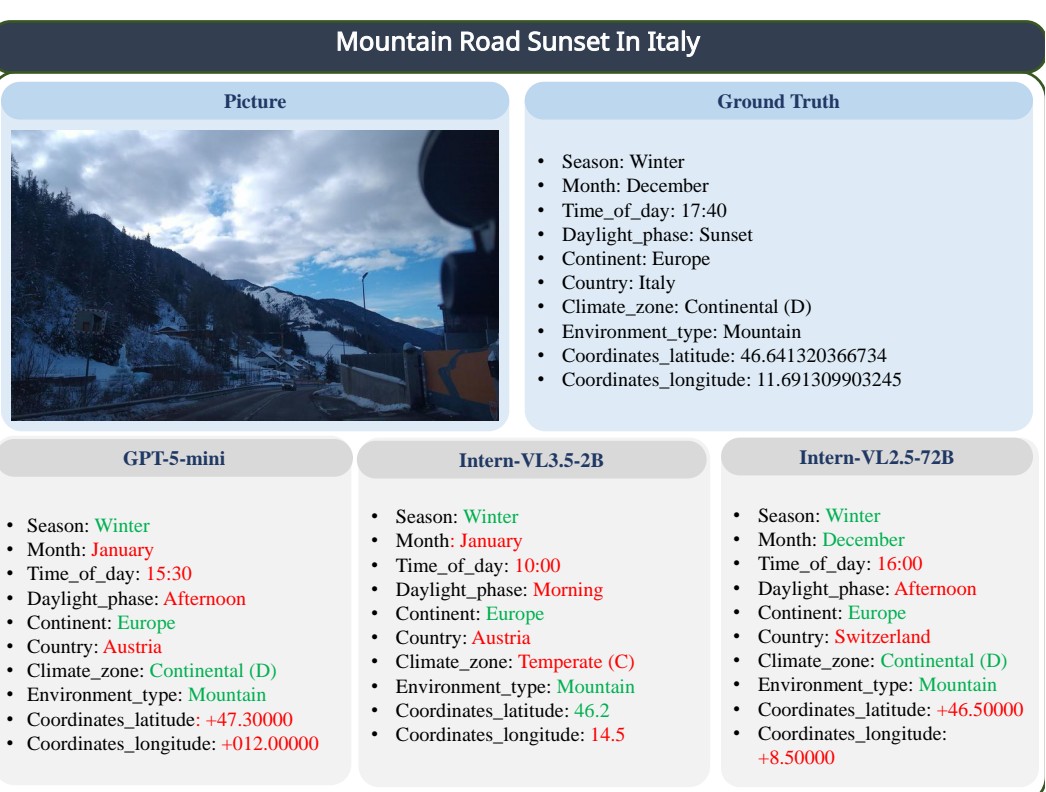

**Figure 28:** Example of TIMESPOT dataset — Mountain Hillside Afternoon In Thailand.

**Figure 29:** Example of TIMESPOT dataset — Mountain Road Sunset In Italy.

## G EXAMPLES OF VERSIONED JSON RECORD

Two such examples from the versioned JSON record are given below:

```
Example : 1
{
    "image_file": "1eda0bc6-d820-440f-8c71-f0289d56de1c.jpg",
    "ground_truth": {
      "season": "Summer",
      "month": "June",
      "time_of_day": "21:36",
      "daylight_phase": "Night",
      "continent": "Europe",
      "country": "Poland",
      "climate_zone": "Continental (D)",
      "environment_type": "Urban",
      "coordinates_latitude": "53.0533603",
      "coordinates_longitude": "16.7307758"
    },
    "raw_model_answer": "- season : Summer\n- month : July\n- time_of_day
        : 11:00\n- daylight_phase : Morning\n- continent : Europe\n-
        country : Poland\n- climate_zone : Continental\n-
        environment_type : Suburban\n- coordinates_latitude : 52.25000\n-
        coordinates_longitude : 19.00000",
    "model_parsed": {
      "season": "Summer",
      "month": "July",
      "time_of_day": "11:00",
      "daylight_phase": "Morning",
      "continent": "Europe",
      "country": "Poland",
      "climate_zone": "Continental",
      "environment_type": "Suburban",
      "coordinates_latitude": "52.25000",
      "coordinates_longitude": "19.00000"
    },
    "judge_output_raw": "json\n{\n  \"season\": {\n    \"ground_truth\":
        \"Summer\",\n    \"model_ans\": \"Summer\",\n    \"evaluation\":
        1\n  },\n  \"month\": {\n    \"ground_truth\": \"June\",\n    \"
        model_ans\": \"July\",\n    \"evaluation\": 0\n  },\n  \"
        time_of_day\": {\n    \"ground_truth\": \"21:36\",\n    \"
        model_ans\": \"11:00\",\n    \"evaluation\": 0\n  },\n  \"
        daylight_phase\": {\n    \"ground_truth\": \"Night\",\n    \"
        model_ans\": \"Morning\",\n    \"evaluation\": 0\n  },\n  \"
        continent\": {\n    \"ground_truth\": \"Europe\",\n    \"
        model_ans\": \"Europe\",\n    \"evaluation\": 1\n  },\n  \"
        country\": {\n    \"ground_truth\": \"Poland\",\n    \"model_ans
        \": \"Poland\",\n    \"evaluation\": 1\n  },\n  \"climate_zone\":
        {\n    \"ground_truth\": \"Continental (D)\",\n    \"model_ans
        \": \"Continental\",\n    \"evaluation\": 1\n  },\n  \"
        environment_type\": {\n    \"ground_truth\": \"Urban\",\n    \"
        model_ans\": \"Suburban\",\n    \"evaluation\": 0\n  },\n  \"
        coordinates_latitude\": {\n    \"ground_truth\": \"53.0533603\",\
        n    \"model_ans\": \"52.25000\",\n    \"evaluation\": 0\n  },\n
        \"coordinates_longitude\": {\n    \"ground_truth\":
        \"16.7307758\",\n    \"model_ans\": \"19.00000\",\n    \"
        evaluation\": 0\n  }\n}\n'''",
    "judge_output_parsed": {
      "season": {
        "ground_truth": "Summer",
        "model_ans": "Summer",
        "evaluation": 1
      },
      "month": {
        "ground_truth": "June",
```

```
        "model_ans": "July",
        "evaluation": 0
      },
      "time_of_day": {
        "ground_truth": "21:36",
        "model_ans": "11:00",
        "evaluation": 0
      },
      "daylight_phase": {
        "ground_truth": "Night",
        "model_ans": "Morning",
        "evaluation": 0
      },
      "continent": {
        "ground_truth": "Europe",
        "model_ans": "Europe",
        "evaluation": 1
      },
      "country": {
        "ground_truth": "Poland",
        "model_ans": "Poland",
        "evaluation": 1
      },
      "climate_zone": {
        "ground_truth": "Continental (D)",
        "model_ans": "Continental",
        "evaluation": 1
      },
      "environment_type": {
        "ground_truth": "Urban",
        "model_ans": "Suburban",
        "evaluation": 0
      },
      "coordinates_latitude": {
        "ground_truth": "53.0533603",
        "model_ans": "52.25000",
        "evaluation": 0
      },
      "coordinates_longitude": {
        "ground_truth": "16.7307758",
        "model_ans": "19.00000",
        "evaluation": 0
      }
    },
    "metrics": {
      "daylight_abs_error_minutes": 636,
      "daylight_abs_error_hm": "10:36",
      "daylight_within_1h": 0,
      "lat_abs_error_deg": 0.8033603000000014,
      "lon_abs_error_deg": 2.2692242,
      "latlon_distance_m": 177222.27432500423
    }
  },

Example 2:
  {
    "image_file": "1eda0bc6-d820-440f-8c71-f0289d56de1c.jpg",
    "ground_truth": {
      "season": "Summer",
      "month": "June",
      "time_of_day": "21:36",
      "daylight_phase": "Night",
      "continent": "Europe",
      "country": "Poland",
      "climate_zone": "Continental (D)",
```

```
            "environment_type": "Urban",
            "coordinates_latitude": "53.0533603",
            "coordinates_longitude": "16.7307758"
        },
        "raw_model_answer": "- season : Summer\n- month : July\n- time_of_day
            : 11:00\n- daylight_phase : Morning\n- continent : Europe\n-
            country : Poland\n- climate_zone : Continental\n-
            environment_type : Suburban\n- coordinates_latitude : 52.25000\n-
            coordinates_longitude : 19.00000",
        "model_parsed": {
          "season": "Summer",
          "month": "July",
          "time_of_day": "11:00",
          "daylight_phase": "Morning",
          "continent": "Europe",
          "country": "Poland",
          "climate_zone": "Continental",
          "environment_type": "Suburban",
          "coordinates_latitude": "52.25000",
          "coordinates_longitude": "19.00000"
        },
        "judge_output_raw": "json\n{\n  \"season\": {\n    \"ground_truth\":
            \"Summer\",\n    \"model_ans\": \"Summer\",\n    \"evaluation\":
            1\n  },\n  \"month\": {\n    \"ground_truth\": \"June\",\n    \"
            model_ans\": \"July\",\n    \"evaluation\": 0\n  },\n  \"
            time_of_day\": {\n    \"ground_truth\": \"21:36\",\n    \"
            model_ans\": \"11:00\",\n    \"evaluation\": 0\n  },\n  \"
            daylight_phase\": {\n    \"ground_truth\": \"Night\",\n    \"
            model_ans\": \"Morning\",\n    \"evaluation\": 0\n  },\n  \"
            continent\": {\n    \"ground_truth\": \"Europe\",\n    \"
            model_ans\": \"Europe\",\n    \"evaluation\": 1\n  },\n  \"
            country\": {\n    \"ground_truth\": \"Poland\",\n    \"model_ans
            \": \"Poland\",\n    \"evaluation\": 1\n  },\n  \"climate_zone\":
             {\n    \"ground_truth\": \"Continental (D)\",\n    \"model_ans
            \": \"Continental\",\n    \"evaluation\": 1\n  },\n  \"
            environment_type\": {\n    \"ground_truth\": \"Urban\",\n    \"
            model_ans\": \"Suburban\",\n    \"evaluation\": 0\n  },\n  \"
            coordinates_latitude\": {\n    \"ground_truth\": \"53.0533603\",\
            n    \"model_ans\": \"52.25000\",\n    \"evaluation\": 0\n  },\n
             \"coordinates_longitude\": {\n    \"ground_truth\":
            \"16.7307758\",\n    \"model_ans\": \"19.00000\",\n    \"
            evaluation\": 0\n  }\n}\n",
        "judge_output_parsed": {
          "season": {
            "ground_truth": "Summer",
            "model_ans": "Summer",
            "evaluation": 1
          },
          "month": {
            "ground_truth": "June",
            "model_ans": "July",
            "evaluation": 0
          },
          "time_of_day": {
            "ground_truth": "21:36",
            "model_ans": "11:00",
            "evaluation": 0
          },
          "daylight_phase": {
            "ground_truth": "Night",
            "model_ans": "Morning",
            "evaluation": 0
          },
          "continent": {
            "ground_truth": "Europe",
```

```
          "model_ans": "Europe",
          "evaluation": 1
        },
        "country": {
          "ground_truth": "Poland",
          "model_ans": "Poland",
          "evaluation": 1
        },
        "climate_zone": {
          "ground_truth": "Continental (D)",
          "model_ans": "Continental",
          "evaluation": 1
        },
        "environment_type": {
          "ground_truth": "Urban",
          "model_ans": "Suburban",
          "evaluation": 0
        },
        "coordinates_latitude": {
          "ground_truth": "53.0533603",
          "model_ans": "52.25000",
          "evaluation": 0
        },
        "coordinates_longitude": {
          "ground_truth": "16.7307758",
          "model_ans": "19.00000",
          "evaluation": 0
        }
      },
      "metrics": {
        "daylight_abs_error_minutes": 636,
        "daylight_abs_error_hm": "10:36",
        "daylight_within_1h": 0,
        "lat_abs_error_deg": 0.8033603000000014,
        "lon_abs_error_deg": 2.2692242,
        "latlon_distance_m": 177222.27432500423
      }
    },
```

