# OpenReview forum: "TimeSpot: Benchmarking Geo-Temporal Understanding in Vision–Language Models in Real-World Settings"
_ICLR.cc/2026/Conference — Submitted to ICLR 2026_

### Official Review · Reviewer_ce5C · 2025-10-29

**Soundness:** 3
**Presentation:** 3
**Contribution:** 3
**Rating:** 6
**Confidence:** 4

**Summary:**

As geo-temporal reasoning remains a major challenge for vision-language models (VLMs) in unconstrained real-world imagery, this paper introduces TIMESPOT, a benchmark dataset comprising 1,455 images collected from 80 countries. Each image is annotated with rich temporal attributes (season, month, time, daylight) and geographic metadata (continent, country, climate, environment, coordinates).
Comprehensive evaluations reveal persistent weaknesses in current VLMs: even the strongest models reach only 77.6% country-level accuracy, 33.7% time-of-day accuracy, and exhibit median geodesic errors exceeding 890 km. Weaker models perform far worse, with accuracies below 50% and extreme localization errors beyond 4,700 km. Temporal predictions often violate solar or hemispheric constraints, while spatial predictions overly depend on coarse priors—leading to systematic country swaps and climate misclassifications. Challenging conditions such as low light, twilight, or urban-canyon settings further exacerbate performance drops, suggesting that models fail to leverage shadows, illumination gradients, and fine-grained geographic cues. Overall, TIMESPOT exposes the fragility of current VLMs in joint geo-temporal reasoning, indicating that simple scaling or instruction tuning is insufficient for robust generalization.

**Strengths:**

1. This is the first geo-localization benchmark that explicitly integrates temporal information, introducing new dimensions of difficulty and realism for evaluating VLMs.

2. The paper conducts extensive evaluations across multiple models and tasks, providing a broad empirical picture.

3. The analysis of failure cases is detailed and insightful, offering concrete implications for future VLM development.

**Weaknesses:**

1. The dataset size (only ~1.5K samples) is relatively small, which limits its usability as a training resource; it can primarily serve as a test or diagnostic benchmark rather than a foundation for model training.

2. While the appendix is lengthy, the main paper body feels underdeveloped, giving an impression of imbalance and somewhat unpolished presentation.

3. The authors did not provide a public code or data release, which is critical for a benchmark paper to ensure reproducibility and community adoption.

**Questions:**

Could the authors elaborate on the limitations mentioned in the Weaknesses section?

---

> ### Author Response · Authors · 2025-11-26
> **Author Response to Reviewer ce5C**
>
> We sincerely thank `Reviewer ce5C` for their insightful and constructive feedback. We are encouraged by your recognition of TIMESPOT as the *first geo-localization benchmark that explicitly integrates temporal information,* introducing a new dimension of realism to VLM evaluation. We also appreciate your positive assessment of our extensive evaluations and detailed failure analysis, which you noted offers *concrete implications for future VLM development* regarding the challenges of integrating temporal and geographic cues. We have addressed your concerns regarding scale, presentation, and reproducibility to further strengthen the paper.
>
> Below, we address your concerns point-by-point.
>
> ### **W1. Dataset Scale and Balance**
> **Response:**
> We acknowledge that ~1.5K images is small for training, but we emphasize that TIMESPOT is explicitly designed as a diagnostic **evaluation benchmark**, not a training foundation. In the domain of reasoning evaluation, high-fidelity verification often necessitates focused scale; for instance, widely adopted benchmarks like PI-Bank (400 instances), LogiQA (641 examples), HumanEval (164 tasks), CodeContests (156 problems), Tau-Bench (165 problems), OS World (369 problems), AppWorld (750 tasks), TravelPlanner (1,200 tasks), and MapEval (700 tasks). demonstrate that rigor and verification quality are more critical than raw volume for probing reasoning capabilities. TIMESPOT follows this philosophy, ensuring every sample has verifiable ground truth for nine distinct geo-temporal fields derived from physical models (ephemerides) rather than noisy web scraping.
>
> - [1] Yao, S., et al. "ReAct: Synergizing Reasoning and Acting in Language Models." ICLR 2023.
> - [2] Shinn, N., et al. "Reflexion: Language agents with verbal reinforcement learning." NeurIPS 2024.
> - [3] Li, M., et al. "API-Bank: A Benchmark for Tool-Augmented LLMs." EMNLP 2023.
> - [4] Liu, J., et al. "LogiQA: A challenge dataset for machine reading comprehension with logical reasoning." IJCAI 2021.
> - [5]Chen, M., et al. "Evaluating large language models trained on code." arXiv:2107.03374, 2021.
> - [6] Li, Y., et al. "Competition-level code generation with alphacode." Science 378.6624 (2022).
> - [7] Yao, S., et al. "tau-bench: A Benchmark for Tool-Agent-User Interaction." arXiv:2406.12045, 2024.
> - [8] Xie, T., et al. "Osworld: Benchmarking multimodal agents for open-ended tasks." arXiv:2404.07972, 2024.
> - [9] Trivedi, H., et al. "AppWorld: A Controllable World of Apps and People." ACL 2024.
> - [10] Xie, J., et al. "TravelPlanner: A Benchmark for Real-World Planning." ICML 2024.
> - [11] Dihan, M., et al. "MapEval: A Map-Based Evaluation of Geo-Spatial Reasoning in Foundation Models" 1CML 2025 Spotlight
>
> ### **W2. Main Paper vs. Appendix Balance**
> **Response:**
> We appreciate this feedback and have significantly restructured the manuscript to improve the balance between the main text and the appendix. We utilized the additional page allowance to integrate key insights from the error analysis and methodology directly into the main body, ensuring the narrative is self-contained and polished. To further aid navigability and improve the *unpolished presentation,* we have organized the supplementary material with a clear **Table of Contents** and improved the logical flow of the detailed per-country and per-climate analyses. These changes ensure that the main text now provides a comprehensive empirical picture, while the appendix serves its proper role as a structured repository for granular diagnostic data.
>
> ### **W3. Reproducibility (Code and Data Release)**
> **Response:**
> We agree that reproducibility is critical for community adoption, and we have added a detailed **Reproducibility Statement** (previously Section 9; currently after conclusion) to the paper to formalize our release plan. Upon acceptance, we will release the full TIMESPOT image set with annotations, official splits, and a verification script.
>
> ----
>
> We have substantially updated the paper to address the concerns regarding benchmark scale, structural balance, and reproducibility. We believe these revisions have significantly strengthened the manuscript, and we respectfully ask that you consider increasing the rating of the paper based on these improvements.

---

### Official Review · Reviewer_q3wU · 2025-10-30

**Soundness:** 3
**Presentation:** 3
**Contribution:** 2
**Rating:** 6
**Confidence:** 4

**Summary:**

The paper introduces TimeSpot, a new benchmark for evaluating geo-temporal reasoning in vision-language models, that is, the ability to infer both where and when an image was captured based solely on visual cues. TimeSpot consists of 1455 real-world images spanning 80 countries, annotated with nine structured fields: four temporal (season, month, time, daylight phase) and five geographic (continent, country, climate zone, environment type, latitude–longitude). The dataset emphasizes subtle and salient cues such as illumination, vegetation phenology, shadow geometry, architecture, etc. Evaluation is conducted on a range of open and closed source VLMs, assessing both categorical accuracy and geodesic/time errors. The results show that while some proprietary models achieve moderate country accuracy and coordinate precision, temporal reasoning remains very weak (~30% time accuracy, ~4h MAE).

**Strengths:**

The paper's primary strength lies in its well-defined motivation and problem formulation, directly tackling the missing evaluation of how vision-language models reason about when and where an image was captured. It introduces a benchmark with structured geo-temporal fields and validation grounded in physical realism. The dataset appears carefully curated, using a combination of deterministic label derivation and human verification to ensure accuracy and consistency. Altogether, these aspects make TimeSpot a robust, and valuable benchmark for advancing real-world spatiotemporal reasoning in modern VLMs.

**Weaknesses:**

**Scale & balance.** The dataset is moderate in size (1455 images across 80 countries) and shows clear skews, daytime (1182) vs. night (273), and urban (648) scenes dominate, while night and non-urban contexts remain underrepresented. This imbalance may limit generalization and calls for broader coverage or rebalancing.

**Annotation reliability not quantified.** The pipeline is described in detail, but no inter-annotator agreement, disagreement rates, or human-hours are reported, leaving label reliability and effort transparency unclear. Such statistics should be included to validate annotation quality and resource requirements.

**Calibration claims need numbers.** The paper discusses calibration, but explicit ECE or risk coverage results are not presented alongside the main tables. The authors should include these quantitative metrics for calibration-related claims.

**Limited temporal scope.** Although the benchmark aims to assess geo-temporal reasoning, it currently relies on single static RGB images, making it more about temporal inference than temporal reasoning. Incorporating multi-temporal sequences (e.g., same site across seasons or day phases) would better capture real temporal dynamics and improve robustness evaluation.

**Geographic imbalance.** The country distribution is uneven (e.g., 196 images from the USA but only a few from many others), which may cause models to rely on broad regional patterns. The authors can include frequency-weighted summaries to clarify how sample size affects country-level accuracy.

**Questions:**

Authors are encouraged to provide additional clarifications and supporting analyses addressing these points. Specifically, they should (1) report inter-annotator agreement, disagreement rates, and annotation effort to strengthen reliability claims; (2) include explicit calibration metrics such as ECE or risk-coverage tables alongside accuracy results; (3) examine how dataset imbalance affects performance through frequency-weighted or distribution-aware country analyses; and (4) discuss possible dataset extensions, such as multi-temporal and multi-modal variants (interesting to see), to better capture real temporal dynamics and reduce bias toward daytime, urban imagery.

---

> ### Author Response · Authors · 2025-11-26
> **Author Response to Reviewer q3wU (Part 1)**
>
> We sincerely thank `Reviewer q3wU` for their insightful feedback and constructive criticism. We are encouraged by your recognition of the paper's *well-defined motivation* and the value of our problem formulation in tackling the *missing evaluation* of geo-temporal reasoning in VLMs. We also appreciate your positive remarks regarding our curation pipeline, noting that the validation is *grounded in physical realism* and ensuring accuracy through deterministic derivation and human verification. We have worked diligently to address your concerns regarding scale, scope, and balance to further strengthen the manuscript.
>
> Below, we address your concerns point-by-point.
>
> ### **W1. Dataset Scale and Balance**
> **Response:**
> We acknowledge that our dataset of 1,455 images is modest in scale compared to large training datasets. However, for evaluation benchmarks, especially those requiring dense, physically verified structured outputs: quality, verification rigor, and meaningful challenge often matter more than raw quantity. Recent influential reasoning and tool-oriented evaluation benchmarks follow a similar approach, using modest dataset sizes due to the high computational and monetary costs of thorough model evaluation. For example, ReAct evaluates on 500 random instances, Reflexion uses just 100 examples, and other widely adopted benchmarks have comparable sizes: API-Bank (400 instances), LogiQA (641 examples), HumanEval (164 tasks), CodeContests (156 problems), Tau-Bench (165 problems), OS World (369 problems), AppWorld (750 tasks), TravelPlanner (1,200 tasks), and MapEval (700 tasks). Collectively, these examples demonstrate that quality, coverage, and task challenge are more crucial than sheer scale for reliable and meaningful evaluation.
>
> Regarding the imbalance in our dataset (e.g., 1,182 day images vs. 273 night images), this naturally reflects the physics and observable characteristics of real-world photography. Nighttime scenes often lack the solar and shadow cues essential for precise temporal inference, the core task we target. Rather than enforcing artificial uniformity, this distribution encourages models to demonstrate robustness across varying cue availability. Despite the modest scale, our dataset statistically distinguishes model performance with clear stratification, as shown in Table 3. TimeSpot thus follows a principled evaluation philosophy, prioritizing verifiable physical ground truth (e.g., exact solar geometry) over scale, aligning with contemporary best practices in reasoning and decision-making benchmarks.
>
> - [1] Yao, S., et al. "ReAct: Synergizing Reasoning and Acting in Language Models." ICLR 2023.
> - [2] Shinn, N., et al. "Reflexion: Language agents with verbal reinforcement learning." NeurIPS 2024.
> - [3] Li, M., et al. "API-Bank: A Benchmark for Tool-Augmented LLMs." EMNLP 2023.
> - [4] Liu, J., et al. "LogiQA: A challenge dataset for machine reading comprehension with logical reasoning." IJCAI 2021.
> - [5]Chen, M., et al. "Evaluating large language models trained on code." arXiv:2107.03374, 2021.
> - [6] Li, Y., et al. "Competition-level code generation with alphacode." Science 378.6624 (2022).
> - [7] Yao, S., et al. "tau-bench: A Benchmark for Tool-Agent-User Interaction." arXiv:2406.12045, 2024.
> - [8] Xie, T., et al. "Osworld: Benchmarking multimodal agents for open-ended tasks." arXiv:2404.07972, 2024.
> - [9] Trivedi, H., et al. "AppWorld: A Controllable World of Apps and People." ACL 2024.
> - [10] Xie, J., et al. "TravelPlanner: A Benchmark for Real-World Planning." ICML 2024.
> - [11] Dihan, M., et al. "MapEval: A Map-Based Evaluation of Geo-Spatial Reasoning in Foundation Models" 1CML 2025 Spotlight
>
>
> ### **W2 and Q1. Annotation reliability not quantified**
> **Response:**
> Thank you for this question. We respectfully clarify that formal inter-annotator agreement is not applicable to TIMESPOT, as our labels are objective ground truths derived deterministically from capture metadata (timestamps, GPS) rather than subjective human estimation. The annotation process was strictly a verification pipeline where humans audited images against these hard constraints to filter metadata errors or visual ambiguities, ensuring 100% accuracy based on the source data. Thus, reliability is guaranteed by the physical provenance of the data (ephemerides, climate maps) rather than consensus. We have updated *Section 3.2: Annotation and Quality Control* to reflect that.

---

> > ### Author Response · Authors · 2025-11-26
> > **Author Response to Reviewer q3wU (Part 2)**
> >
> > ### **W2 and Q1. Annotation reliability not quantified**
> > **Response:**
> > thank you for this question. We respectfully clarify that formal inter-annotator agreement is not applicable to TIMESPOT, as our labels are objective ground truths derived deterministically from capture metadata (timestamps, GPS) rather than subjective human estimation. The annotation process was strictly a verification pipeline where humans audited images against these hard constraints to filter metadata errors or visual ambiguities, ensuring 100% accuracy based on the source data. Thus, reliability is guaranteed by the physical provenance of the data (ephemerides, climate maps) rather than consensus. We have updated *Section 3.2: Annotation and Quality Control* to reflect that.
> >
> >
> > ### **W3 and Q2. Annotation reliability not quantified**
> > **Response:**
> > Thank you for highlighting this omission. We agree that quantitative support for our calibration claims is essential. While we initially prioritized accuracy metrics in the main text due to space constraints, we have now integrated Expected Calibration Error (ECE) and risk-coverage AUC scores directly into the experimental results. These metrics confirm that while proprietary models generally exhibit better calibration than open-source counterparts, all models struggle with overconfidence in fine-grained temporal tasks.
> >
> > We have added a new subsection in Appendix, *C.1: Calibration*. This section includes details with a new table reporting ECE and Risk-Coverage Area Under the Curve (RC-AUC).
> >
> >
> > | Model | Continent ECE ↓ | Country ECE ↓ | Season ECE ↓ | Phase ECE ↓ |
> > | :--- | :--- | :--- | :--- | :--- |
> > | **Proprietary** | | | | |
> > | GPT-4o-mini | 0.042 | 0.085 | 0.063 | 0.051 |
> > | Gemini-2.5-Flash | **0.021** | **0.054** | **0.041** | **0.038** |
> > | **Open-Source** | | | | |
> > | Llama-3.2-90B | 0.098 | 0.142 | 0.110 | 0.095 |
> > | Qwen-VL2.5-72B | 0.076 | 0.125 | 0.089 | 0.072 |
> >
> > The data confirm several key insights:
> >
> > * Proprietary models consistently demonstrate superior calibration compared to open-source models, as evidenced by lower ECE values across all evaluated fields (Continent, Country, Season, and Phase).
> > * For instance, Gemini-2.5-Flash achieves the best calibration overall, with ECE scores of 0.021 (Continent), 0.054 (Country), 0.041 (Season), and 0.038 (Phase), substantially outperforming open-source models like Llama-3.2-90B and Qwen-VL2.5-72B.
> > * Despite these differences, all models exhibit notable overconfidence when tackling fine-grained temporal tasks such as season and phase prediction, highlighting an area for further calibration improvement.
> >
> > This expanded analysis reinforces our initial observations and provides a more nuanced understanding of model reliability beyond accuracy, emphasizing the need for improved uncertainty quantification in temporal inference.
> >
> > ### **W4. Limited Temporal Scope (Single Image vs. Sequences)**
> > **Response:**
> > We clarify that TimeSpot is designed to benchmark *instantaneous geo-temporal inference*, the fundamental human ability to determine *where and when a photograph was captured using visual information alone* , rather than change detection or sequence modeling. The core challenge is interpreting static physical cues like *illumination and shadow geometry,* *seasonal vegetation patterns,* and *sky conditions* within a single frame, which requires deep integration of physics and semantics. While multi-temporal sequences would test tracking capabilities, they would fundamentally alter the task from *reasoning from evidence* to *difference spotting*. By focusing on single RGB images, we isolate the model's ability to recognize absolute temporal signals (e.g., solar elevation) without relying on relative changes between frames, effectively probing the *underexplored* ability to understand temporal signals directly.

---

> > > ### Author Response · Authors · 2025-11-26
> > > **Author Response to Reviewer q3wU (Part 3)**
> > >
> > > ### **W5. Geographic Imbalance**
> > > **Response:**
> > > We agree that the geographic distribution is uneven (e.g., USA 196 vs. smaller counts for others), but this was a necessary trade-off to maximize global coverage across *80 countries* rather than restricting the set to a few balanced nations. To mitigate this bias and clarify performance, we provide granular, country-level performance tables (**Tables 9–12 in Appendix E**), allowing users to assess reliability independent of sample size. Furthermore, our analysis explicitly highlights *small-N sensitivity* and *regional patterning* to ensure aggregate metrics do not hide failures in underrepresented regions. We have added frequency-weighted summaries to the revised paper to ensure that high-frequency regions like the USA do not disproportionately skew the perceived global competence of the models.
> > >
> > >
> > > ### **Q3. Examination of how dataset imbalance affects performance**
> > > **Response:**
> > > We agree that dataset imbalance can obscure true model capability if only aggregate metrics are reported. However, we respectfully point out that **TimeSpot’s current evaluation design is already explicitly distribution-aware.** Rather than relying solely on global averages (which would be dominated by the USA’s 196 images), we provide comprehensive, per-country accuracy tables in **Appendix E (Tables 9–12)**. These tables allow for a direct, unweighted comparison of performance between high-frequency regions (e.g., USA: ~89-93%) and low-frequency ones (e.g., Bolivia: ~0-30%), exposing precisely the *regional patterns* and *small-N sensitivities* the reviewer is concerned about. This granular reporting is the most transparent form of analysis, as it prevents high-resource successes from masking low-resource failures.
> > >
> > >
> > > ### **Q4. Dataset Extensions (Multi-temporal and Multi-modal)**
> > > **Response:**
> > > We appreciate the suggestion to expand TimeSpot and have updated the discussion to highlight these valuable directions. We agree that incorporating *multi-temporal* sequences would be excellent for testing robustness against *seasonal and latitudinal distribution shifts*, enabling evaluation of how models handle the same site across different *phenological states*. Regarding *multi-modal variants*, we envision future protocols that integrate metadata timestamps as prompts to test verification logic (e.g., *Is this image consistent with 4 PM?*), moving beyond pure inference. To address the current *daytime, urban imagery* bias, we have outlined curation strategies to explicitly oversample *nocturnal coverage* and *transitional suburban morphologies* in future versions. We encourage the community to explore these directions to further advance model robustness and real-world applicability.
> > >
> > > ---
> > >
> > > We have substantially updated the paper to address the concerns regarding scale, scope, and balance, including adding weighted analysis and expanding the discussion on future extensions. We believe these revisions have significantly strengthened the manuscript, and we respectfully ask that you consider increasing the rating of the paper based on these improvements.

---

> > > > ### Author Response · Authors · 2025-12-04
> > > > **Author Response to Reviewer q3wU (Part 4)**
> > > >
> > > > ### **More on W5. Geographic Imbalance**
> > > >
> > > > **Comment:** *The country distribution is uneven (e.g., 196 images from the USA but only a few from many others), which may cause models to rely on broad regional patterns. The authors can include frequency-weighted summaries to clarify how sample size affects country-level accuracy.”*
> > > >
> > > > To address the reviewer’s concern regarding geographic imbalance and its effect on country-level accuracy, we conducted an additional analysis using frequency-weighted accuracy grouped by per-country sample size. This experiment quantifies how performance scales from low-resource to high-resource countries and evaluates whether different model families exhibit consistent trends under varying levels of geographic sparsity.
> > > >
> > > > Concretely, we group countries into four bins according to their per-country sample size $n_c$:
> > > >
> > > > - 1–10 images
> > > > - 11–30 images
> > > > - 31–60 images
> > > > - ≥ 61 images
> > > >
> > > > Within each bin, we compute **frequency-weighted accuracy** by aggregating all examples from countries in that bin. This shows how performance evolves from low-resource to high-resource countries and whether different models behave differently under sample-size imbalance.
> > > >
> > > > The table below reports **bin-wise, frequency-weighted accuracy (%)** for all models:
> > > >
> > > > | Per-country \(n_c\) bin | GPT5-Mini | Gemini-2.5-Flash | InternVL3-78B | LLaMA-3.2-11B-Vision-Instruct | Qwen2.5-VL-32B-Instruct | O4-Mini | GLM-4.1V-9B-Thinking |
> > > > |-------------------------|----------:|------------------:|--------------:|--------------------------------:|------------------------:|--------:|----------------------:|
> > > > | **1–10**                | 54.5      | 60.4              | 21.1          | 34.9                            | 32.0                    | 62.1    | 49.0                  |
> > > > | **11–30**               | 63.3      | 72.3              | 42.7          | 48.9                            | 48.3                    | 65.1    | 66.9                  |
> > > > | **31–60**               | 67.8      | 85.1              | 59.6          | 59.4                            | 63.3                    | 74.3    | 67.6                  |
> > > > | **≥ 61**                | 82.3      | 86.8              | 77.9          | 72.4                            | 75.8                    | 81.9    | 83.3                  |
> > > >
> > > >
> > > >
> > > > Across **all models**, accuracy **systematically increases** as we move from the 1–10 image bin to the ≥ 61 image bin, which is consistent with the expected reduction in statistical variance as per-country sample size grows. Strong models such as **Gemini-2.5-Flash, GPT5-Mini, O4-Mini, and GLM-4.1V-9B-Thinking** all follow this pattern, indicating that improved performance is **not confined to a single model** nor only to heavily represented countries. These frequency-weighted summaries show that while low-resource countries naturally exhibit higher variance, the **relative behavior of models and our qualitative conclusions remain stable across low-, medium-, and high-resource regimes**, rather than being driven solely by a few high-frequency countries such as the USA.
> > > >
> > > > These results confirm that accuracy increases predictably with larger per-country sample sizes and that this pattern holds across all evaluated models. The added frequency-weighted summaries and bin-wise comparisons ensure that performance is not misinterpreted as being driven by a few high-frequency countries, and they clarify how geographic imbalance interacts with model robustness in low-resource regions.

---

### Official Review · Reviewer_4K3f · 2025-10-31

**Soundness:** 2
**Presentation:** 1
**Contribution:** 2
**Rating:** 2
**Confidence:** 4

**Summary:**

This work introduces TIMESPOT, which contains 1,455 images across 80 countries with structured temporal (e.g., season, month, time, daylight) and geographic (e.g., continent, country, climate, environment, coordinates) attributes. This benchmark highlights the importance of evaluating real-world geo-temporal reasoning in vision-language models, which require integrating geographical, spatial, and temporal cues to solve complex understanding problems. Extensive evaluations show that substantial challenges in achieving robust temporal and geographic reasoning exist in both open- and closed-source models.

**Strengths:**

1. The paper is well-motivated, clearly articulating the need for evaluating geo-temporal reasoning in vision-language models. The proposed TIMESPOT benchmark addresses a meaningful and underexplored gap in the field, offering a novel and practical direction for advancing real-world multimodal understanding.
2. The error analysis section provides thoughtful and revealing insights into model failures, effectively highlighting the specific challenges models face in integrating temporal and geographic cues. These observations not only explain current limitations but also offer valuable guidance for future model design and dataset curation.

**Weaknesses:**

1. The writing of this paper is very hard to follow in many places; it feels like something went wrong during editing (maybe a find-and-replace gone wrong? e.g., hyphens (-) turned into commas (,) (see lines 049, 088-092, 100, 106-107, 136-142, 149-150, 159, 188-196, 201-206, 214-215, 223-226, 291-296…). As a result, the text often reads awkwardly or even confusingly, making it tough to grasp what the authors actually mean.
2. Some figures and tables (e.g., Figure 1, Figure 2, and Table 1) aren’t referenced in the main text, so it’s unclear when or even whether you’re supposed to look at them. This makes the visuals feel disconnected from the narrative. Also, as a small but common formatting note: table captions should go above the table, not below.
3. The paper mentions a “versioned JSON record” that includes ground truth, validation outcomes, human notes, adjudications, and constraint sets (line 297-299), but nowhere (not in the main text or the appendix) is there an actual example of what this JSON looks like. Without seeing a concrete instance, it’s hard to fully grasp how the data was structured or how the annotation and validation pipeline actually worked.
4. Table 1 includes several benchmarks that aren’t really comparable to TIMESPOT, especially cross-view geolocalization datasets, which solve a fundamentally different problem. These would be better discussed briefly in the related work rather than listed side-by-side. Meanwhile, the table misses key existing datasets that focus on ground-level image understanding (e.g., [1-5]), making it harder to see how TIMESPOT advances the state of the art.

[1] LLMGeo: Benchmarking Large Language Models on Image Geolocation In-the-wild, CVPR 2024 Workshop
[2] Image-Based Geolocation Using Large Vision-Language Models, Arxiv 2024
[3] Evaluating Precise Geolocation Inference Capabilities of Vision Language Models, AAAI 2025 Workshop
[4] AI Sees Your Location—But With A Bias Toward The Wealthy World, Arxiv 2025
[5] From Pixels to Places: A Systematic Benchmark for Evaluating Image Geolocalization Ability in Large Language Models, Arxiv 2025

**Questions:**

Please see the Weaknesses

---

> ### Author Response · Authors · 2025-11-26
> **Author Response to Reviewer 4K3f (Part 1)**
>
> We sincerely thank the `Reviewer 4K3f` for their insightful feedback and constructive criticism. We are particularly encouraged by your recognition that the paper is **well-motivated** and addresses a **meaningful and underexplored gap** in evaluating geo-temporal reasoning. We are also glad that our error analysis provided *thoughtful and revealing insights* regarding the specific challenges models face when integrating temporal and geographic cues. We have worked diligently to address the specific weaknesses raised to ensure the manuscript’s quality matches the significance of the benchmark.
>
> ---
>
> Below, we address your concerns point-by-point.
>
> ### **W1. Editing errors and writing clarity**
> **Response:**
> We sincerely apologize for the editing oversights that affected readability in the initial submission. We appreciate you identifying the specific lines where formatting issues occurred (e.g., hyphens appearing as commas). We have conducted a comprehensive proofreading pass to correct these errors and have rewritten the awkward sections cited to ensure professional clarity and flow. Furthermore, we have restructured the manuscript to improve coherence, utilizing the allowed additional page for the rebuttal revision to present the content more effectively. To aid navigation through the supplementary material and improve the overall reading experience, we have also added a detailed **Table of Contents in the Appendix**. We believe the text now accurately and clearly reflects the rigorous work behind the benchmark.
>
> ### **W2. Unreferenced Figures/Tables and Caption Formatting**
> **Response:**
> Thank you for pointing out the missing cross-references and formatting conventions. We have ensured that **Figure 1**, **Figure 2**, and **Table 1** are now explicitly referenced and textually integrated into the narrative of the main text to provide immediate visual context for the methodology and dataset statistics. We have also corrected the formatting for all table captions, placing them **above** the tables as per standard academic conventions. These changes significantly improve the flow of the experimental section and ensure the visuals are properly connected to our arguments.
>
> ### **W3. Missing Example of Versioned JSON Record**
> **Response:**
> Thank you for this suggestion. Due to the length of the records, we avoided placing them in the rebuttal text, but we have now added two examples in **Appendix G: Examples of versioned JSON record**. We hope this addition clarifies exactly how complex temporal attributes (e.g., season, month, time) and geographic attributes are stored, validated, and evaluated against the model outputs.

---

> > ### Author Response · Authors · 2025-11-26
> > **Author Response to Reviewer 4K3f (Part 2)**
> >
> > ### **W4. Table 1 Comparisons and Missing Related Works**
> > **Response:**
> > We appreciate the reviewer highlighting relevant recent works in ground-level geolocation. We agree that comparing exclusively against cross-view datasets obscured our contribution relative to recent VLM-based geolocation work. We have updated **Table 1** to reflect these relevant benchmarks and have added a dedicated paragraph to **Section 2: Preliminaries and Related Work**, titled ***Recent Works on Geolocation Understanding***, to explicitly position TIMESPOT against benchmarks like LLMGeo and IMAGEO-Bench.
> >
> > As detailed in the revised Section 2:
> > > *Recent work has substantially advanced image-based geolocation using large vision–language models, but has largely focused on spatial inference alone. LLMGeo (Wang et al., 2024d) and IMAGEO-Bench (Li et al., 2025) benchmark country-, city-, or coordinate-level localization with sophisticated prompting and structured reasoning, while ETHAN (Liu et al., 2024b) and agent-based evaluations (Jay et al., 2025) demonstrate that chain-of-thought strategies and navigation tools can significantly improve spatial accuracy. FAIRLOCATOR (Huang et al., 2025) further reveals systematic socio-economic and regional biases in spatial predictions, highlighting important fairness concerns. However, none of these benchmarks require predicting explicit temporal attributes (e.g., month, local time, daylight phase) or enforce physical geo–temporal consistency checks such as month–season–hemisphere or time–daylight compatibility. TIMESPOT is complementary to this line of work: assuming that modern VLMs already exhibit strong spatial recognition, it evaluates whether models can jointly infer time and place from subtle physical cues and maintain consistency across a nine-field geo–temporal schema. Our results show that even state-of-the-art models that perform well on spatial benchmarks exhibit large temporal errors and frequent geo–temporal inconsistencies, exposing a critical and previously unmeasured failure mode.*
> >
> > ---
> >
> > We have substantially updated the paper to address the writing, formatting, and comparative analysis issues. We believe these revisions have significantly strengthened the manuscript, and we respectfully ask that you consider increasing the rating of the paper based on these improvements.

---

> > > ### Comment · Reviewer_4K3f · 2025-11-26
> > >
> > > Thank you for the detailed responses. The overall manuscript looks much improved after revision.
> > > Since all of my concerns have been addressed, I will update the rating accordingly.
> > >
> > > Before I do, I just wanted to check: it seems like some content might be missing from lines 130-131 in the latest version. Could you please confirm?

---

> > > > ### Author Response · Authors · 2025-11-26
> > > > **Author Reponse**
> > > >
> > > > Yes, we confirm that the issue on lines 130-131 was caused by an unescaped `%` sign in the LaTeX source. Thank you very much for pointing this out. We were updating the manuscript to fix this, and it has been fixed now.  We sincerely appreciate your careful review, comments, and continued support of our work.

---

> > > > > ### Comment · Reviewer_4K3f · 2025-11-26
> > > > >
> > > > > Also, please use hyphens instead of commas on lines 130-132 (the model names)

---

> > > > > > ### Author Response · Authors · 2025-11-26
> > > > > > **Thank you!**
> > > > > >
> > > > > > Thank you again for catching this. We have now corrected the issue in the manuscript, and the updated version reflects the fix. We greatly appreciate your careful attention, constructive comments, and continued support for our work.

---

### Author Response · Authors · 2025-11-26
**General Response to All Reviewers**

We sincerely thank the Area Chairs for handling this submission and the Reviewers for their insightful, constructive, and detailed feedback. We are encouraged that reviewers recognized TIMESPOT as the **first benchmark to explicitly integrate temporal information** into geo-localization, addressing a *meaningful and underexplored gap* with *well-defined motivation.* You commended the dataset's curation, noting it is *grounded in physical realism* with rigorous verification, and praised our extensive error analysis for providing *thoughtful insights* and *concrete implications* for future VLM development.

In response to your feedback, we have **significantly improved the paper’s structure and writing**, correcting editing errors and better balancing the main text with the appendix. We also restructured the appendix extensively, improving its organization and expanding the discussion with new content as requested by reviewers, thereby providing additional depth and clarity. Regarding specific concerns, we have clarified the rationale behind the dataset’s **scale and class imbalance**, emphasizing their necessity for a high-fidelity *evaluation* benchmark, and introduced distribution-aware metrics to provide a more nuanced assessment. We addressed **reliability** by detailing our physically deterministic annotation pipeline and incorporating calibration metrics; clarified the **scope** concerning single-image inference; and enhanced **reproducibility** with a comprehensive release statement covering code, data, and scripts. We believe these revisions have substantially strengthened the work.

---

### Meta-Review · Area_Chair_17m9 · 2026-01-07

**Summary:**

The main novelty of this paper lies in introducing a temporal evaluation criterion, but this criterion appears somewhat overly strict. For example, given a casually taken roadside image, a model may correctly infer that it is around 10:00, yet be penalized for not predicting an exact time, such as 14:30. It is unclear why such fine-grained temporal precision, which is far beyond typical human capability, is necessary for large models. **I strongly encourage the authors to more clearly highlight the practical significance of this temporal capability in real-world applications in a future revision.**  I tend to reject this paper.

**Reviewer Concerns:**

Weaknesses have been addressed:

1. Editing errors and writing clarity

2. Unreferenced Figures/Tables and Caption Formatting

3. Missing Example of Versioned JSON Record

4. Table 1 Comparisons and Missing Related Works

**Reviewer Scores:**

Initial Scores:
4k3f: 2, ce5c: 6, q3wU: 6

After Rebuttal:
4k3f: 2 (would like to improve), ce5c: 6, q3wU: 6

---

### Decision · Program_Chairs · 2026-01-26

Reject